# Debugging Concept Bottleneck Models through Removal and Retraining

**Eric Enouen**
Cornell University
enouen@cs.cornell.edu

**Sainyam Galhotra**
Cornell University
sg@cs.cornell.edu

## Abstract

Concept Bottleneck Models (CBMs) use a set of human-interpretable concepts to predict the final task label, enabling domain experts to not only validate the CBM's predictions, but also intervene on incorrect concepts at test time. However, these interventions fail to address systemic misalignment between the CBM and the expert's reasoning, such as when the model learns shortcuts from biased data. To address this, we present a general interpretable debugging framework for CBMs that follows a two-step process of *Removal* and *Retraining*. In the *Removal* step, experts use concept explanations to identify and remove any undesired concepts. In the *Retraining* step, we introduce `CBDebug`, a novel method that leverages the interpretability of CBMs as a bridge for converting concept-level user feedback into sample-level auxiliary labels. These labels are then used to apply supervised bias mitigation and targeted augmentation, reducing the model's reliance on undesired concepts. We evaluate our framework with both real and automated expert feedback, and find that `CBDebug` significantly outperforms prior retraining methods across multiple CBM architectures (PIP-Net, Post-hoc CBM) and benchmarks with known spurious correlations[1].

## 1 Introduction

Concept Bottleneck Models (CBMs) (Koh et al., 2020) have emerged as a powerful architecture for interpretable vision classification. A CBM consists of two stages: a concept extractor first predicts a set of human-understandable concepts, which are then passed to an inference layer to produce the final label. This intermediate representation allows domain experts to inspect the model's reasoning process and verify whether it aligns with their own. This capability is crucial in high-stakes domains, such as healthcare or scientific analysis, where errors are costly (Makary & Daniel, 2016) and expert validation is essential (Lekadir et al., 2025).

Beyond passive validation, CBMs enable test-time interventions (Koh et al., 2020; Espinosa Zarlenga et al., 2023). An expert can review the predicted concepts and directly correct them to influence the final prediction. For example, if a radiologist corrects a mispredicted concept, such as marking a lesion as present when the model missed it, the diagnosis may shift from benign to malignant. Such interventions elevate the expert from a passive auditor to an active participant in decision making. However, the effectiveness of these interventions hinges on a critical assumption: the learned concepts must align with expert knowledge. In practice, this alignment is often fragile. Data quality issues, sampling bias, and incomplete concept vocabularies can lead models to exploit spurious correlations or overlook key factors (Torralba & Efros, 2011; Geirhos et al., 2020). Because test-time edits address only surface errors, the same reasoning flaws inevitably reappear on new samples.

Existing approaches fall short of achieving reliable alignment and are prone to such issues. Supervised CBMs attempt to enforce alignment by sharing a concept vocabulary with annotators, but they require costly per-sample labels and remain vulnerable to concept leakage, which can obscure global misalignment (Havasi et al., 2022; Srivastava et al., 2024). Unsupervised CBMs (Chen et al., 2019; Debole et al., 2025) reduce labeling demands by discovering concepts from data or leveraging foundation models, yet this flexibility increases the risk that the learned concept set diverges from expert understanding.

---

[1] Code available at https://github.com/ericenouen/cbdebug

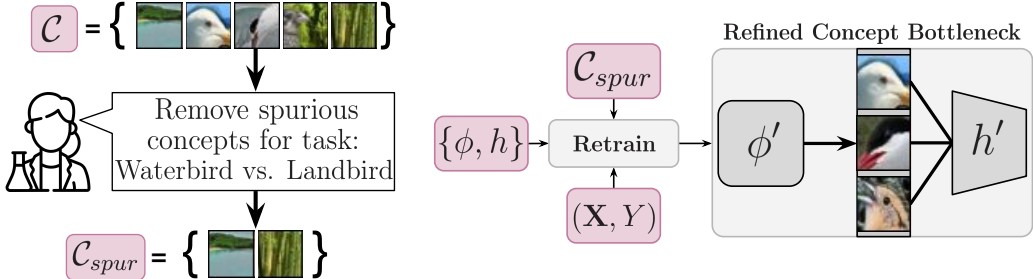

(a) Removal. User removes undesired concepts $\mathcal{C}_{spur}$.

(b) Retraining. Update encoder $\phi$ and simple layer $h$ based on $\mathcal{C}_{spur}$.

Figure 1: Our debugging framework for incorporating a domain expert's knowledge into a concept bottleneck. **Removal (a)**: The user inspects concept explanations and selects undesired concepts to remove, such as background concepts in bird classification. **Retraining (b)**: The concept extractor and inference layer are retrained based on this feedback, updating the CBM to remove dependence on undesired concepts while maintaining reliance on task-relevant concepts.

In this work, we present a general debugging framework for CBMs that enables experts to globally edit a model's reliance on undesired concepts, ensuring that its predictions are not only accurate but also *right for the right reasons* (Ross et al., 2017), and aligned with the domain expert's reasoning. This framework follows a two-step process of **Removal** and **Retraining** (Figure 1).

In the Removal step (Figure 1a), experts evaluate concept explanations and eliminate those spuriously correlated with the label. For example, an ornithologist may remove background concepts unrelated to bird species (Rao et al., 2024). However, removal alone is insufficient: remaining concepts may still carry signals from the removed ones, and task-relevant concepts may have been ignored in favor of spurious ones. To address this, we introduce a Retraining step (Figure 1b) that leverages expert feedback to guide the model toward an expert-aligned concept set.

To implement our retraining step, we propose `CBDebug` (Concept Bottleneck Debugger), which operationalizes interpretable debugging by treating expert feedback as a causal intervention. `CBDebug` leverages the interpretability of CBMs as a bridge to first convert the expert feedback into sample-level auxiliary labels. Then, using the estimated auxiliary labels, performs a reweighting and augmentation scheme to approximate the counterfactual distribution where the undesired concepts have no effect on the label. In summary, we make the following core contributions:

- We present an interpretable debugging framework for CBMs, extending to a more general architecture and enabling domain experts to globally edit model reasoning.
- We introduce `CBDebug`, a retraining approach that first approximates sample-level auxiliary labels from concept-level feedback, then reweights and augments the dataset to reduce reliance on undesired concepts and better align the model with expert reasoning.
- We validate our framework across multiple CBMs (PIP-Net, Post-hoc CBM) and datasets with known spurious correlations. `CBDebug` most effectively leverages user feedback on spurious concepts, outperforming prior work on ProtoPNets and improving worst-group accuracy by up to 26% over the original model, with strong results when feedback is automated with an LLM.

## 2 RELATED WORK

We review prior work on concept bottleneck models, interpretable debugging, and bias mitigation, and position our approach at their intersection.

**Concept Bottleneck Architecture.** Concept bottleneck models (CBMs) (Koh et al., 2020) decompose prediction into a concept extraction stage and an inference stage. While supervised CBMs require concept annotations to ensure alignment with human-defined attributes, such labels are costly to obtain and cannot be assumed for spurious concepts. We focus on recent unsupervised CBMs, which learn concepts directly from data and allow experts to discover and address undesired shortcuts.

These models fall into two main architectural families. *ProtoPNets* (Chen et al., 2019) and their extensions (Nauta et al., 2023b; Ma et al., 2024; Carmichael et al., 2024), learn prototypical patches from the training set to make predictions. In this family, concept-level explanations take the form of representative image patches from the training data. *VLM-CBMs* (Debole et al., 2025) use a CLIP backbone to score the presence of textual concepts, as explored in many approaches (Yuksekgonul et al., 2023; Oikarinen et al., 2023; Srivastava et al., 2024), where concept-level explanations are the textual descriptions of each concept. Our framework can also extend to post-hoc XAI methods (e.g., interpreting neurons from class activation maps (Zhou et al., 2016) as concepts). In this work we focus on concept bottlenecks, which are interpretable-by-design, and evaluate one representative from each family: PIP-Net (Nauta et al., 2023b) and Post-hoc CBM (Yuksekgonul et al., 2023).

**Interpretable Debugging.** The goal of interpretable debugging is to enable a domain expert to interact with an interpretable model to detect and correct undesired behaviors. Early work focused on explanatory interactive learning (Ross et al., 2017; Teso & Kersting, 2019; Schramowski et al., 2020), with subsequent extensions to neuro-symbolic models (Stammer et al., 2022; 2021). Bontempelli et al. (2023) advanced these approaches, focusing on *ProtoPNets* and removing the need for a fixed concept vocabulary. Building on top of these works, we focus on a generalized unsupervised CBM architecture (Poeta et al., 2023).

Within unsupervised CBMs, spurious concept removal has been explored in both *ProtoPNets* and *VLM-CBMs*: Nauta et al. (2023a) study removal in PIP-Net on medical datasets, while Yuksekgonul et al. (2023); Rao et al. (2024); Chauhan et al. (2023) evaluate similar strategies for *VLM-CBMs*. Retraining efforts have focused mainly on *ProtoPNets*, such as adding object segmentation maps for extra supervision (Barnett et al., 2021) or ProtoPDebug (Bontempelli et al., 2023) which applies a forgetting loss on class-specific subsets of user-identified spurious concepts. Donnelly et al. (2025) targets task-specific spurious concepts and bypasses the need for retraining, but is limited to random patch selection for learning new concepts. For *VLM-CBMs*, Bontempelli et al. (2021) outline debugging strategies but lack empirical evaluation, while Hu et al. (2024) supports concept removal without the need for retraining through influence functions. We build on these directions with a general debugging framework for CBMs and an effective retraining approach, `CBDebug`, that leverages a novel connection between interpretable debugging and bias mitigation to remove undesired task-specific concepts and better align models with expert feedback.

**Bias Mitigation.** Interpretable debugging and bias mitigation are related but distinct. Debugging aligns a model with an expert's reasoning through explicit feedback, while bias mitigation aims to improve robustness by reducing reliance on spurious correlations. We further explore connections to two main groups of bias mitigation: *supervised methods*, which incorporate auxiliary labels, and *unsupervised methods*, which estimate spurious correlations directly from data.

*Supervised methods* require auxiliary labels to reduce the impact of spurious correlations (Zheng & Makar, 2022; Sagawa et al., 2019; Makar et al., 2022; Kirichenko et al., 2022). While these approaches cannot be directly applied to unsupervised CBMs, `CBDebug` utilizes a crucial connection between interpretable debugging and these methods. By leveraging the model's interpretability, we can bridge concept-level human feedback to sample-level auxiliary labels, effectively removing the reliance on any concept marked as undesired by the domain expert. This is accomplished by collecting the activation scores for each concept on every sample, which can then be used to form the auxiliary labels. Specifically, we instantiate our approach with permutation weighting (Arbour et al., 2021), which can handle high-dimensional, real-valued auxiliary labels. This allows us to use concept activations directly, unlike methods such as GroupDRO (Sagawa et al., 2019) that require discrete groups and would necessitate an additional clustering step to convert activation scores.

*Unsupervised methods* have similarly been proposed that relax the requirement for auxiliary labels. These approaches either automatically estimate spurious groups (Sohoni et al., 2020; Seo et al., 2022; Chakraborty et al., 2024) or reweight samples based on assumptions about how spurious correlations are learned during training (Liu et al., 2021; Nam et al., 2020; Espinosa Zarlenga et al., 2024; Wu et al., 2023). Instead of relying on underlying assumptions about model training dynamics, our method focuses only on removing concepts marked directly by a domain expert. While the two approaches can overlap, interpretable debugging offers a distinct and complementary advantage: it gives the expert fine-grained control over what should be removed. For example, an expert may wish to keep certain 'spurious' concepts if they know they will perform well in practice, or may wish to remove 'core' concepts to debug what the model would use instead. This ensures the resulting model

is aligned with the domain knowledge of the expert, which is critical for interpretable models that are part of a human-machine team. Further discussion and results can be found in Appendix C.4.

# 3 CONCEPT BOTTLENECK DEBUGGING FRAMEWORK

In this section, we formalize our interpretable debugging framework, for leveraging expert feedback to eliminate undesired concepts and aligning the model's reasoning with the expert's preferences. We first define the general class of concept bottleneck models our framework supports. Then, we outline our two-step debugging process: how concept-level user feedback is collected during the removal step and the formal goal of the retraining step.

## 3.1 CONCEPT BOTTLENECK

We denote a concept bottleneck model as a pair $\{\phi, h\}$. The concept extractor $\phi : \mathcal{X} \to \mathbb{R}^m$ maps an input $x \in \mathcal{X}$ to a vector of $m$ concept activation scores, and the inference layer $h : \mathbb{R}^m \to \mathcal{Y}$, typically a sparse linear layer, maps these activations to the output label. The core requirement for a concept bottleneck is that each concept has a corresponding human-interpretable explanation, and we review recent work that falls under this definition in Section 2 (*Concept Bottleneck Architecture*).

Specifically, we focus on unsupervised CBMs, which eliminate the need for auxiliary labels and are thus applicable to a broader range of real-world settings. These models enable automatic concept discovery, making them scalable, but they are also prone to learning concepts that are entirely spurious or irrelevant from the perspective of a domain expert. Our framework is particularly well-suited to address this challenge, as it provides a mechanism for experts to inspect, debug, and guide the concepts learned by these scalable models. Furthermore, standard CBMs are not immune to shortcut learning: unsupervised CBMs simply make these shortcuts more explicit, rather than hidden among other concepts, enabling users to identify and remove them.

## 3.2 REMOVAL

We focus on the removal of concepts that encode biases undesirable for the classification task, as identified by the domain expert. For instance, when classifying birds, the expert may wish to remove confounded concepts that capture background information rather than features of the birds themselves. Our framework does not assume a specific structure or representation for the underlying concepts. Instead, it operates in a general setting where each concept is associated with an explanation, allowing our method to be applied across a variety of concept discovery approaches.

To guide concept removal, we adopt a simple binary feedback mechanism that assumes spuriosity is defined at the task level: each concept is either retained or marked for removal, based on expert input. This minimal supervision design ensures that our approach remains broadly applicable and easy to integrate into real-world workflows, where experts may have limited time or domain knowledge to provide detailed annotations, but does not directly account for class-specific spurious concepts.

We illustrate our Removal process in Figure 1a. We assume a trained CBM $\{\phi, h\}$ with learned concept set $\mathcal{C} = \{c_1, \ldots, c_m\}$. A domain expert inspects the learned concept set by interacting with the concept explanations and identifies a subset $\mathcal{C}_{spur} \subset \mathcal{C}$ to remove. We remove all concepts in $\mathcal{C}_{spur}$ from the concept set. Then, we pass the edited CBM $\{\phi, h\}$ and $\mathcal{C}_{spur}$ to the retraining step.

## 3.3 RETRAINING

There are two main failure modes when removing $\mathcal{C}_{spur}$ from $\mathcal{C}$. First, if any remaining concepts partially encode information from the undesired concepts, removal alone may not fully eliminate their influence. Second, if the model relied too heavily on the undesired concepts, removal can leave the model unable to perform well. These limitations motivate the need for retraining, which adapts the CBM to maintain high performance while avoiding reliance on the marked undesired concepts.

Having obtained human feedback in the form of a set of concepts $\mathcal{C}_{spur}$ deemed spurious for the task, we face a crucial challenge: **how can we most effectively use this feedback to improve the model's reasoning and performance?**

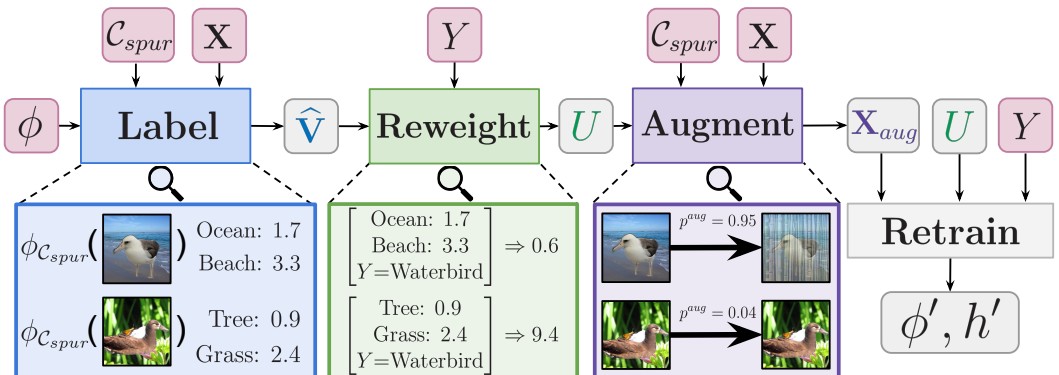

Figure 2: Overview of `CBDebug` (Concept Bottleneck Debugger), which consists of three main steps. First, the encoder $\phi$ computes the concept activations for undesired concepts in $\mathcal{C}_{spur}$ to generate the approximated auxiliary label $\widehat{\mathbf{V}}$. Second, permutation weighting utilizes $\widehat{\mathbf{V}}$ and the class label $Y$ to compute the odds of the sample being drawn from the unconfounded distribution, generating weights $\mathbf{U}$. Third, augmentation is performed on $\mathbf{X}$ based on the undesired concepts $\mathcal{C}_{spur}$ and weights $\mathbf{U}$ to generate $\mathbf{X}_{aug}$. Finally, we retrain $\{\phi, h\}$ on $(\mathbf{X}_{aug}, Y)$ weighted by $\mathbf{U}$ and return $\{\phi', h'\}$.

We illustrate our Retraining process in Figure 1b. The retraining algorithm is given the trained concept bottleneck $\{\phi, h\}$, the training dataset $(\mathbf{X}, Y)$, and the set of undesired concepts $\mathcal{C}_{spur}$. The goal of this step is to return an updated concept bottleneck $\{\phi', h'\}$ that maintains high task performance by leveraging other, more task-relevant concepts instead of $\mathcal{C}_{spur}$.

## 4 CBDEBUG: CONCEPT BOTTLENECK DEBUGGER

To achieve this goal, we introduce `CBDebug` (Figure 2), which operationalizes interpretable debugging by treating expert feedback as a causal intervention. Intuitively, `CBDebug` treats undesired concepts as observed confounders and approximates the counterfactual distribution where those confounders have no effect on the label. This human-in-the-loop approach allows experts to explicitly select or refine the set of undesired concepts, giving them a transparent mechanism to steer the model, unlike methods that rely on unsupervised group discovery (see **Bias Mitigation** in Section 2).

In practice, `CBDebug` consists of three main stages:

- **Label** (Section 4.1): Convert concept-level feedback into sample-level auxiliary labels by scoring each sample with the CBM. The activation scores $\widehat{\mathbf{V}}$ approximate the true auxiliary labels $\mathbf{V}$ for undesired concepts.
- **Reweight** (Section 4.2): Apply permutation weighting (Arbour et al., 2021) on $(\widehat{\mathbf{V}}, Y)$ to compute sample weights $\mathbf{U}$ that approximate the unconfounded distribution.
- **Augment** (Section 4.3): Use the sample weights $\mathbf{U}$ to selectively augment bias-aligned samples, yielding a dataset $\mathbf{X}_{aug}$ that further reduces shortcut reliance.

We then fine-tune the concept bottleneck with the augmented dataset $(\mathbf{X}_{aug}, Y)$ weighted by sample weights $\mathbf{U}$ and return the refined concept bottleneck $\{\phi', h'\}$.

### 4.1 LABEL

The first stage of our approach (**Label** in Figure 2) generates sample-level auxiliary labels from the expert's feedback. For example, a user may mark background concepts like 'beach' or 'grass' as undesired for the task of classifying birds. We then utilize the trained CBM's concept extractor to collect the activation scores for the marked concepts on the entire training dataset.

Formally, the labeling step takes the trained concept extractor $\phi$, the training samples $\mathbf{X}$, and the spurious concept set $\mathcal{C}_{spur}$ as input. It returns $\widehat{\mathbf{V}}$, a matrix of dimension $N \times |\mathcal{C}_{spur}|$, where $N$ is the number of samples.

$$\widehat{\mathbf{V}} = \left[ \phi_{\mathcal{C}_{spur}}(x_i) \right]_{i=1}^{n}$$

where $\phi(x_i)$ are the concept activation scores of all concepts in $\mathcal{C}$ for sample $x_i$, and $\phi_{\mathcal{C}_{spur}}(x_i)$ denotes the subselection of those concept activation scores for only the concepts in $\mathcal{C}_{spur}$.

## 4.2 REWEIGHT

The second stage of our approach (**Reweight** in Figure 2) utilizes a supervised bias mitigation approach to reweight the training dataset to reduce the correlation between $\widehat{\mathbf{V}}$ and $Y$. For example, if backgrounds spuriously correlate with class label, the reweighting scheme may assign a low weight to a waterbird on a 'beach' background and a high weight to a waterbird on a 'grass' background, forcing the model to learn features that generalize beyond the undesired background concept.

We adopt permutation weighting (Arbour et al., 2021), later applied to shortcut removal by Zheng & Makar (2022), to perform reweighting. Unlike group-based reweighting approaches such as GroupDRO (Sagawa et al., 2019), which require discrete group labels and often struggle with groups that have low support (necessitating clustering of $\widehat{\mathbf{V}}$), permutation weighting naturally accommodates multi-dimensional, continuous auxiliary labels. By directly enforcing independence between $\widehat{\mathbf{V}}$ and $Y$, it provides a more general and stable mechanism.

Formally, the reweighting step takes the approximated auxiliary labels $\widehat{\mathbf{V}}$ and the class labels $Y$ as input. It returns sample weights $\mathbf{U}$.

Given $\widehat{\mathbf{V}}$ and $Y$, we first construct two datasets. A dataset $D$ is constructed as $\widehat{\mathbf{V}}$ concatenated with $Y$ representing the confounded distribution, where there exists a correlation between the label $Y$ and auxiliary label $\widehat{\mathbf{V}}$. Then, we create a new dataset $D'$ by randomly permuting the label $Y$ in the original dataset. This naturally breaks any correlation between $Y$ and $\widehat{\mathbf{V}}$, representing the unconfounded distribution.

We then train a binary predictor $\eta : Y \times \widehat{\mathbf{V}} \to \{0, 1\}$ to predict the probability of a sample belonging to the unconfounded dataset $D'$ compared to the confounded dataset. Finally, we compute a weight $u_i$ for each sample

$$u_i = \frac{\eta(y_i, v_i)}{1 - \eta(y_i, v_i)} \tag{1}$$

where $\eta(y_i, v_i)$ denotes the estimated probability that a sample belongs to $D'$. To ensure robust weights, we perform K-fold cross validation and average over multiple permutations. See Appendix C.1 for further analysis of effect of the number of folds and number of permutations on assigned sample weights per subgroup.

## 4.3 AUGMENT

The third stage of our approach (**Augment** in Figure 2) aims to further reduce the correlation between $\widehat{\mathbf{V}}$ and $Y$ through augmentation. While reweighting is effective, it can lead to unstable training when the spurious groups are highly imbalanced, as a few samples are given very large weights. Augmentation offers a more robust way to mitigate bias in these scenarios by generating new samples for underrepresented groups. For example, we augment the image of a waterbird on a water background with an image of bamboo from the concept bank, while leaving the image of a waterbird on a land background untouched.

Formally, the augmentation step takes the training samples $\mathbf{X}$, sample weights $\mathbf{U}$, and the concept set $\mathcal{C}_{spur}$ as input. It returns new training samples $\mathbf{X}_{aug}$ that further reduce the correlation between undesired concepts and the class label. Importantly, because these concepts were explicitly marked as undesired by the user, we do not change the label $Y$.

Samples assigned a low weight are more likely to be aligned with the bias we want to remove, and so we would like to focus our augmentation on these samples. To accomplish this, we can convert each sample weight $u_i$ into an augmentation probability $p_{aug}(x_i)$. We first invert the sample weight $u_i$ by subtracting each weight from the maximum sample weight assigned. Then we normalize the resulting values to $[0, 1]$ to convert them into probabilities, and raise the probabilities to a power $\gamma$ to increase contrast and reduce the likelihood of augmenting useful samples. See AppendixC.3 for further analysis on the impact of $\gamma$ on augmentation probabilities and downstream performance.

Table 1: Average and Worst-Group Accuracy on MetaShift and Waterbirds with PIP-Net and Post-hoc CBM. Best in **bold**, second best underlined. Average and standard deviation reported over the three initial seeds for Original and over the six debugging sessions for removal and all retraining approaches. CBDebug consistently improves worst-group accuracy across models and datasets.

| Method | PIP-Net | | | | Post-hoc CBM | | | |
| --- | --- | --- | --- | --- | --- | --- | --- | --- |
| | Waterbirds | | MetaShift | | Waterbirds | | MetaShift | |
| | Average | Worst | Average | Worst | Average | Worst | Average | Worst |
| Original | $92.3_{\pm0.3}$ | $71.9_{\pm2.7}$ | $80.9_{\pm1.3}$ | $52.4_{\pm2.0}$ | $63.5_{\pm1.3}$ | $25.8_{\pm3.0}$ | $92.9_{\pm0.4}$ | $84.5_{\pm2.2}$ |
| Remove | $92.6_{\pm0.4}$ | $74.4_{\pm2.2}$ | $81.4_{\pm0.6}$ | $55.0_{\pm2.6}$ | $61.2_{\pm18.8}$ | $13.9_{\pm15.8}$ | $89.0_{\pm4.8}$ | $73.9_{\pm15.0}$ |
| Retrain | $92.4_{\pm0.1}$ | $72.5_{\pm1.0}$ | $81.2_{\pm1.6}$ | $53.3_{\pm2.1}$ | $66.9_{\pm2.8}$ | $33.2_{\pm6.4}$ | $93.1_{\pm0.7}$ | $84.4_{\pm2.7}$ |
| ProtoPDebug | $92.5_{\pm0.1}$ | $71.6_{\pm1.9}$ | $80.9_{\pm1.4}$ | $52.4_{\pm1.4}$ | - | - | - | - |
| *Ours* | | | | | | | | |
| Reweight Only | $93.2_{\pm0.4}$ | $74.2_{\pm4.8}$ | $81.8_{\pm1.4}$ | $\underline{56.1}_{\pm1.3}$ | $80.0_{\pm8.0}$ | $\mathbf{55.6}_{\pm15.2}$ | $93.1_{\pm0.4}$ | $\underline{87.3}_{\pm1.8}$ |
| Augment Only | $92.4_{\pm0.6}$ | $\underline{75.5}_{\pm2.9}$ | $82.2_{\pm1.7}$ | $55.6_{\pm3.3}$ | $64.5_{\pm4.8}$ | $25.9_{\pm11.4}$ | $92.6_{\pm1.7}$ | $86.3_{\pm4.5}$ |
| **CBDebug** | $93.7_{\pm0.7}$ | $\mathbf{79.4}_{\pm4.3}$ | $82.3_{\pm1.7}$ | $\mathbf{57.3}_{\pm3.1}$ | $73.6_{\pm6.3}$ | $\underline{51.9}_{\pm16.2}$ | $93.4_{\pm1.0}$ | $\mathbf{89.3}_{\pm1.3}$ |

We then augment each sample with probability $p_{aug}(x_i)$. Our augmentation strategy depends on the concept representation:

- *ProtoPNets:* Since ProtoPNets operate on prototypical patches, we first randomly select $k$ spurious concepts from $\mathcal{C}_{spur}$ and perform CutMix (Yun et al., 2019) for each selected concept. To determine the actual patch used, we randomly choose from the top-ten highest-activated prototypical patches for that concept, following Nauta et al. (2023b)'s approach for visualizing concepts.
- *VLM-CBMs:* Since VLM-CBMs use text-based concepts, we leverage a text-to-image-generated concept bank, following DISC (Wu et al., 2023). We randomly select a single spurious concept from $\mathcal{C}_{spur}$ and perform Mixup (Zhang, 2017) with an image of that concept randomly sampled from the concept bank.

This approach ensures that bias-aligned samples are more likely to be augmented, creating a richer training distribution that reduces spurious correlations without altering the true labels.

## 5 EXPERIMENTS

To evaluate our approach, we aim to answer the following questions: **Q1:** Quantitatively, how does `CBDebug` perform on both real (Section 5.1) and automated (Section 5.2) feedback sources? **Q2:** Qualitatively, does `CBDebug` effectively remove dependence on undesired concepts and lead to a more robust concept set (Section 5.3)? We also explore additional ablations of our method and comparisons to unsupervised baselines in Appendix C.

**Datasets.** We use datasets with known spurious correlations: Waterbirds (Sagawa et al., 2019), MetaShift (Liang & Zou, 2022), CelebA (Liu et al., 2015), and ISIC (Codella et al., 2019). These datasets provide concrete testbeds for assessing how well `CBDebug` reduces reliance on undesired concepts, as their group structures allow performance to be measured directly across subpopulations.

**Models.** We evaluate a representative *ProtoPNet* (PIP-Net (Nauta et al., 2023b)) and *VLM-CBM* (Post-hoc CBM (Yuksekgonul et al., 2023)) on these datasets. PIP-Net uses a ConvNeXt-tiny backbone, while Post-hoc CBM uses a CLIP-ViT-L-14 backbone for all datasets except ISIC where BioMedCLIP (Zhang et al., 2023) is used, and both are trained following the authors' original implementation. For Post-hoc CBM, we use a combination of synthetic concepts from Wu et al. (2023) and curated high-quality concepts following Oikarinen et al. (2023) (Appendix B). Each model is trained with three random seeds per dataset, and we report average-group and worst-group accuracy averaged across seeds (see Appendix A.1 for additional training details). For ISIC we follow Wu et al. (2023) and report test AUROC since there are $2^7$ distinct groups.

**Setup.** After training the original models, we collect feedback from a real or automated domain expert to identify spurious concepts (Appendix A.2). Since the expert feedback is aligned with the known spurious correlations, we utilize the robustness of the model as a measure for the effectiveness of each retraining algorithm. For PIP-Net, we fine-tune the entire model for half the original training epochs. For Post-hoc CBM, we freeze the backbone and retrain only the linear layer.

Table 2: Average and Worst-Group Accuracy for Automated Feedback on Post-hoc CBM. Average and standard deviation reported over the three initial seeds. `CBDebug` consistently outperforms the original model and standard retraining.

| Method | Waterbirds | | MetaShift | | CelebA | | ISIC |
|---|---|---|---|---|---|---|---|
| | Average | Worst | Average | Worst | Average | Worst | AUROC |
| Original | $63.5_{\pm1.3}$ | $25.8_{\pm3.0}$ | $92.9_{\pm0.4}$ | $\underline{84.5}_{\pm2.2}$ | $76.2_{\pm0.8}$ | $8.7_{\pm0.9}$ | $39.3_{\pm3.7}$ |
| Remove | $64.6_{\pm20.7}$ | $2.5_{\pm1.1}$ | $90.5_{\pm4.6}$ | $79.6_{\pm12.7}$ | $19.9_{\pm9.1}$ | $6.5_{\pm9.1}$ | $41.7_{\pm16.9}$ |
| Retrain | $69.0_{\pm2.2}$ | $38.0_{\pm5.5}$ | $92.4_{\pm0.5}$ | $83.0_{\pm2.2}$ | $79.9_{\pm0.9}$ | $22.2_{\pm5.9}$ | $37.7_{\pm5.9}$ |
| *Ours* | | | | | | | |
| Reweight Only | $80.1_{\pm10.0}$ | $\mathbf{61.9}_{\pm15.8}$ | $92.0_{\pm1.8}$ | $84.1_{\pm5.2}$ | $73.9_{\pm5.4}$ | $\mathbf{53.3}_{\pm5.3}$ | $\underline{52.6}_{\pm5.2}$ |
| Augment Only | $67.4_{\pm2.7}$ | $32.9_{\pm6.7}$ | $92.0_{\pm1.5}$ | $84.4_{\pm4.8}$ | $71.5_{\pm6.2}$ | $38.9_{\pm12.6}$ | $18.6_{\pm8.1}$ |
| **CBDebug** | $76.0_{\pm2.8}$ | $\underline{58.3}_{\pm6.0}$ | $93.0_{\pm1.7}$ | $\mathbf{87.5}_{\pm2.8}$ | $68.7_{\pm4.1}$ | $\underline{51.3}_{\pm3.9}$ | $\mathbf{58.0}_{\pm11.6}$ |

**Baselines.** We compare `CBDebug` against the following baselines (Appendix A.3):

- Removal. Removes undesired concepts without further retraining.
- Retraining. Takes the model after removal and fine-tunes it on the training dataset, following a standard fine-tuning protocol.
- ProtoPDebug (Bontempelli et al., 2023). Collects image patches in input space representing undesired concepts into a forget set, penalizes the encoder for activating on forget set patches.
- Reweight/Augment Only. These ablations evaluate our main components in isolation: the Label and Reweight step (without augmentation) and the Augment step (without reweighting).

## 5.1    CAN CBDEBUG EFFECTIVELY RETRAIN BASED ON USER FEEDBACK?

To answer this question, we run debugging sessions with six real users. Each user performed the removal step for each of the four dataset-model combinations evaluated in Table 1 (Appendix A.2). For both models, users are instructed to select spurious concepts. For PIP-Net, users are shown the top ten most activated patches from the training dataset and optionally three example images showing which patch the concept activates on. For Post-hoc CBM, users are shown the full set of learned concepts and they can select concepts that seem spurious for the task. We fine-tune according to each user's feedback on each model, making each session an end-to-end debugging run.

**Baselines**. Our results are shown in Table 1. For PIP-Net, our removal baseline provides a modest boost to worst-group accuracy, improving performance by 2.5% on Waterbirds and 2.6% on MetaShift. In contrast, for Post-hoc CBM, removal substantially reduces worst-group accuracy. We hypothesize that this is because Post-hoc CBM's more limited concept set (roughly 10-30) causes it to ignore other task-relevant concepts in favor of the dominant shortcut, making it more sensitive to removing bias-aligned concepts than PIP-Net, which learns far more concepts (around 100-200) (Appendix A.2). For both models, there remains a significant gap between the worst-group and average-group accuracy, indicating that spurious correlations were not fully eliminated. While concept removal can yield incremental improvements, it cannot by itself encourage the model to discover new, robust concepts and is insufficient for fully addressing shortcut reliance.

The Retrain baseline further illustrates this point. For PIP-Net, it performs worse than Removal, while for Post-hoc CBM, it improves performance on Waterbirds but not on MetaShift. These results suggest that even when spurious concepts are explicitly removed, retraining on the biased dataset can cause the same correlations to leak back into the model's representations, highlighting the need for a more targeted retraining approach.

**CBDebug**. In contrast to these naive baselines, `CBDebug` improves worst-group accuracy by 7.5% on Waterbirds and 4.9% on MetaShift for PIP-Net, and by 26.1% on Waterbirds and 4.8% on MetaShift for Post-hoc CBM. `CBDebug` surpasses the previous state-of-the-art interpretable debugger, ProtoPDebug, while integrating its component steps into a framework that delivers more stable gains across settings.

Table 3: Top five concepts for Post-hoc CBM before retraining, after retraining normally, and after retraining with `CBDebug`. Retrain learns new background concepts (highlighted in blue and green) to replace the ones removed. `CBDebug` effectively removes background concepts, replacing them with more robust concepts.

| Class | Original | Retrain | CBDebug |
|---|---|---|---|
| Waterbird | hooked seabird beak
sea
harbor
lake
gull-like body | beach
gull-like body
water
hooked seabird beak
duck-like body | duck-like body
hooked seabird beak
orange wings
orange eyes
orange nape |
| Landbird | olive crown
tree-clinging-like body
forest
olive upper tail
tree | olive upper tail
bamboo
green primary color
tree-clinging-like body
olive breast | olive upper tail
iridescent bill
blue upper tail
olive crown
hawk-like body |

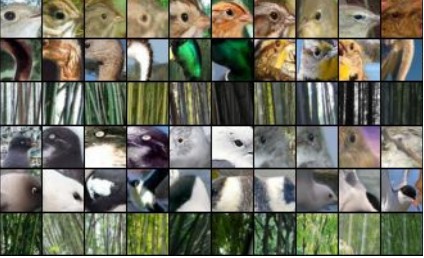 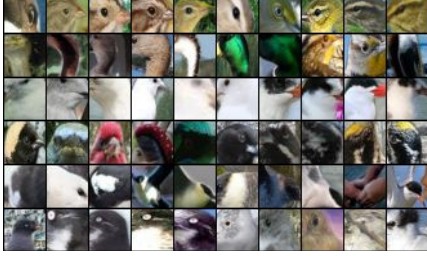

(a) Concepts before retraining      (b) Concepts after retraining with `CBDebug`

Figure 3: The six most highly activated concepts for the Original model trained on Waterbirds and the model after retraining with `CBDebug`. `CBDebug` removes both concepts representing bamboo from the concept set and replaces them with more robust concepts representing bird features.

These consistent improvements across architectures and datasets highlight `CBDebug` as a reliable and effective method for debugging based on real user feedback. We additionally compare `CBDebug` to unsupervised group-robustness methods in Appendix C.4, which rely on assumptions about how spurious correlations are learned, showing the benefit of human-guided debugging for performing bias mitigation.

## 5.2 CAN CBDEBUG EFFECTIVELY RETRAIN BASED ON AUTOMATED FEEDBACK?

As our framework incorporates a domain expert in the loop, a natural question is whether this feedback can be automated with recent advances in foundation models. To explore this, we use LLMs to provide automated feedback for Post-hoc CBM on Waterbirds and MetaShift. Automation reduces both human effort and cost, enabling us to further extend experiments to CelebA (Liu et al., 2015) and ISIC (Codella et al., 2019). For Post-hoc CBM, the automated "user" provides a binary judgment on the spuriosity of each text-based concept. We set the temperature to zero to ensure deterministic responses. Additional details on the prompt design and comparisons to real user feedback can be found in Appendix A.2.

**Baselines**. As shown in Table 2, while the Reweight Only baseline achieves superior worst-group performance on Waterbirds and CelebA, its results are less stable across datasets, underperforming on MetaShift. Similarly, the Removal method proves highly volatile on ISIC and MetaShift, where it occasionally performs well but frequently collapses below the original model's performance.

**CBDebug**. In contrast, `CBDebug` offers a more reliable and robust solution, consistently outperforming the original model across all tested benchmarks, with gains of up to 42.6% over the original model on CelebA.

### 5.3 Does CBDebug effectively remove dependence on undesired concepts?

We visualize the concept bottleneck before and after retraining with `CBDebug` on Waterbirds to better understand the impact of our approach on the concept set. For Post-hoc CBM, results in Table 3 show that while baseline retraining still finds new spurious correlations to replace the removed ones, `CBDebug` effectively removes these concepts from the representation and replaces them with task-relevant concepts. Similarly, Figure 3 shows that for PIP-Net, two land concepts that previously dominated the predictions were effectively removed and replaced with more robust bird concepts. In both scenarios, `CBDebug` effectively removes dependence on spurious attributes. Additional visualizations are provided in Appendix C.2.

## 6 Conclusions

We address the problem of misalignment between a model's behavior and domain expert reasoning, often caused by shortcuts from biased data. We propose a general interpretable debugging framework and introduce `CBDebug`, which leverages the interpretability of the model to convert high-level concept feedback into sample-level labels. Empirical results show that `CBDebug` outperforms prior retraining methods across multiple CBMs, datasets, and both real and automated feedback sources.

## 7 Limitations

While CBDebug demonstrates consistent improvements across datasets and model architectures, several limitations remain. First, we observe substantial variance in performance metrics, particularly when using Post-hoc CBM as the backbone, where results are highly sensitive to initialization and training stochasticity. Although CBDebug reliably improves performance across settings, the effect sizes should be interpreted with this variability in mind.

Second, CBDebug's effectiveness depends on the quality and consistency of concept feedback: inaccurate or adversarial feedback can undermine performance gains. Finally, our framework assumes task-level spurious concepts and therefore does not directly mitigate class-specific spurious correlations, which may limit applicability in settings where spuriosity varies across classes.

## 8 Acknowledgements

We thank Linxi Zhao and Sofian Zalouk for insightful discussions on the ideas and manuscript. This research was supported by a gift to the LinkedIn–Cornell Bowers Strategic Partnership, ARO grant W911NF-25-1-0254, BSF grant 2024101 and a grant from Infosys. Any opinions, findings, and conclusions or recommendations expressed in this material are those of the authors and do not necessarily reflect those of the sponsors.

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

# A EXPERIMENTAL DETAILS

## A.1 INITIAL TRAINING

We evaluate two models, each representative of a major family of concept bottlenecks:

- **PIP-Net** (Nauta et al., 2023b): A patch-based concept bottleneck that learns prototypes from the training data using self-supervised losses to make predictions.
- **Post-hoc CBM** (Yuksekgonul et al., 2023): A text-based concept bottleneck that maps the embedding space of a model to similarity scores to textual concepts with concept activation vectors or CLIP embeddings.

We evaluate these two models on three popular subpopulation shift benchmarks:

- **Waterbirds** (Sagawa et al., 2019): A synthetic dataset where the background (water vs. land) acts as the spurious attribute. It contains 4795 training samples.
- **MetaShift** (Liang & Zou, 2022): A natural dataset where the spurious attribute is the scene context (indoor vs. outdoor). We use the version derived from the COCO dataset rather than Visual Genome, containing 2738 training samples.
- **CelebA** (Liu et al., 2015): A face attribute dataset where gender (female vs. male) serves as the spurious attribute. It includes 162770 training samples.
- **ISIC** (Codella et al., 2019): A medical imaging dataset for skin lesion classification into benign or malignant. The spurious attributes are 'dark corners', 'hair', 'gel borders', 'gel bubbles', 'ruler', 'ink markings/staining', and 'patches'. We follow the setup from DISC (Wu et al., 2023) with five different splits testing reliance on spurious features.

All experiments were conducted on a compute node with 112 CPU cores, 1 TB of RAM, and 2×NVIDIA RTX 6000 Ada GPUs. In the next three sections, we explain the experimental details for the components of our approach. Section A.1 describes how we train the original models, Section A.2 describes our user or automated debugging of the original models, and Section A.3 describes our retraining approaches based on the user feedback.

In our experiments we first train each model with three random seeds. For both models we utilize the hyperparameters recommended in their work.

For PIP-Net, we utilize a ConvNeXt-tiny (Liu et al., 2022) backbone. We pretrain for 10 epochs, then train in the second stage for 60 epochs. We utilize a batch size of 128 for pretraining and 64 for training and a learning rate of 0.0005 for the backbone, and 0.05 for the linear layer. For Post-hoc CBM, we use a CLIP-ViT-L-14 backbone. The first step is to initialize a concept bank, which we describe in detail in Section B. We utilize a $\lambda_{\text{sparse}} = 0.02$ for all datasets.

## A.2 USER DEBUGGING SESSIONS

In this section we detail our user study. We run six different debugging sessions with computer science graduate students. Each debugging session consisted of the user marking concepts as spurious on four different tasks: {Waterbirds + PIP-Net, Waterbirds + Post-hoc CBM, MetaShift + PIP-Net, MetaShift + Post-hoc CBM}.

We first show the task description given to study participants, and then provide examples of the user interface for selecting concepts as spurious or not.

Before beginning our small-scale user study, we sought IRB guidance. As the first step, we contacted the IRB office affiliated with the authors' institution, providing a description of our planned study design to determine the appropriate next steps. The compliance assistant responded that, because the research focused on the debugging method and did not involve collecting any user information, "I can confirm based on the information you've provided that we would not consider this project to meet the regulatory definition of human participant research, and therefore you do not need to submit an application to conduct the work as you have described it." Based on this determination, we did not proceed with a formal application.

Participants were computer science graduate students who voluntarily chose to take part in the debugging sessions. The study was not part of a course requirement, and participation was not tied

to grades, credit, or other obligations. No personal data or sensitive information was collected, and the activities involved brief, task-focused feedback on visual model explanations. Participation was entirely voluntary, and no compensation was provided. Participants were informed of the study's purpose and that their contributions would be used in a research paper.

---

**User Study Task Description**

In this study, you will help improve two state-of-the-art interpretable vision classification models: PIP-Net and Post-hoc CBM. These models aim to explain their predictions using human-understandable concepts.

However, these interpretable models still suffer from shortcut learning, where they latch on to spurious correlations that do not hold robustly in the real world. A classic example is a model trained to recognize wolves that mistakenly learns to associate the presence of snow in the background with the wolf class, because most training images of wolves happened to include snowy scenes.

In this study, we give you the opportunity to improve these models by identifying and removing such spurious concepts.

MODELS

- PIP-Net learns visual concepts. You will be shown visual features the model has identified as important. Each concept has both its top-10 image patches visualized as well as three images where this prototype is marked as active that can be optionally viewed. **Mark concepts that do not focus on the correct object for the classification task.**

- Post-hoc CBM uses text-based concepts. We've seeded its concept bank with some potentially spurious candidates in addition to the core concepts, and you'll see which concepts the model relied on. **Mark those that seem irrelevant or non-causal for the prediction.**

DATASETS AND TASKS

You will perform this analysis across three datasets:

- Waterbirds – Classify images as either a waterbird (e.g. Albatross, Auklet, Gull) or landbird (e.g. Woodpecker, Hummingbird, Warbler).

- MetaShift – Classify images as either a dog or cat.

TASK DETAILS

For each dataset, you will:

1. Review the concepts learned by each model and how they relate to the prediction labels.

2. Flag any concepts you believe are misleading, spurious, or unrelated to the class being predicted.

You will repeat this process for both models. Your input will help teach the model which concepts to unlearn to build a more robust concept set.

---

We also automate the feedback for Post-hoc CBM with a large language model, GPT-3.5-turbo. We show below the task description prompt used, with the specific classification_task_description dependent on the dataset being used.

---

**Automated User Study Task Description**

You are a helpful assistant that classifies visual concepts as either SPURIOUS or NOT SPURIOUS.
The classification task is: {classification_task_description}
A concept is considered SPURIOUS if:
1. It is NOT a physical or anatomical attribute of the object itself.
2. It may correlate with the label due to dataset bias (e.g., background scenery or co-occurring objects), but is not causally related to the object's identity.

Respond only with SPURIOUS or NOT SPURIOUS and a brief justification.

---

**Classification Task Descriptions**

- Waterbirds: "distinguish between WATERBIRDS and LANDBIRDS."
- MetaShift: "distinguish between common animal categories such as CATS and DOGS."
- CelebA: "distinguish between people with BLONDE HAIR and DARK HAIR."

---

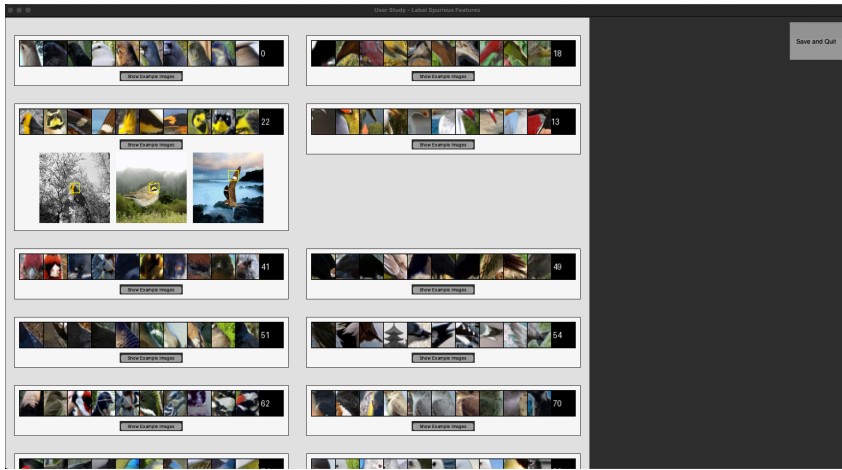

(a) Example of our user interface for patch-based models

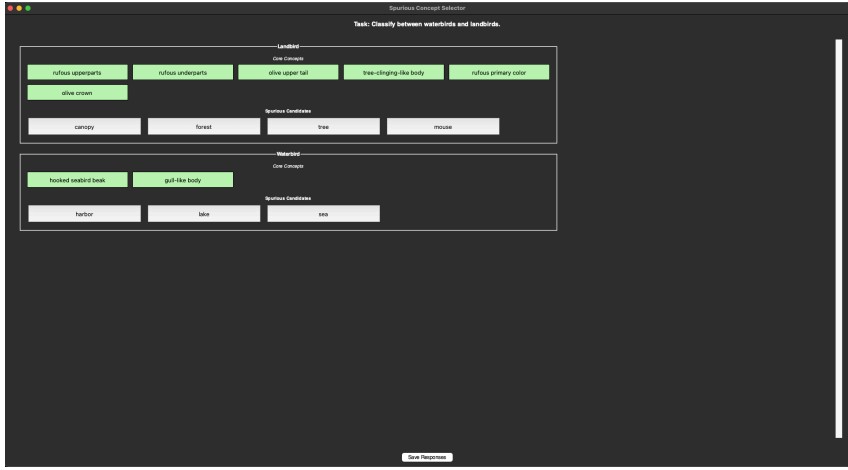

(b) Example of our user interface for text-based models

Figure 4: Participants are shown concepts learned by the model and asked to flag those that are spurious for the classification task.

We then show the number of initial concepts used by the model compared to the number removed by the users. For the user results, we average the initial concepts over the random three seeds, the removed concepts over the six debugging sessions. For the automated results, we average over the three random seeds.

For PIP-Net (Figure 5), we see fairly consistent results across the six users and three random seeds, showing that users generally agree on which concepts are spurious. Since we have two users annotating each model, we can also compute the average agreement. On Waterbirds the average agreement is 97.9%, and on MetaShift the average agreement is 82.4%.

For Post-hoc CBM (Figure 6), the users again seem to remove around the same number of concepts, although the agreement scores vary much more with average agreement on Waterbirds being 51.4% and average agreement on MetaShift being 44.8%. However, we point out that even though the agreement is not high, retraining can still work well across users as we show in our main results.

Finally we also show our automated results on Post-hoc CBM (Figure 7), showing that it removes more concepts on average than the real users.

Additionally, while we focus on automating text-based models in this work, multi-modal models could be utilized to extend the automated results to patch-based models and we leave exploration of this to future work.

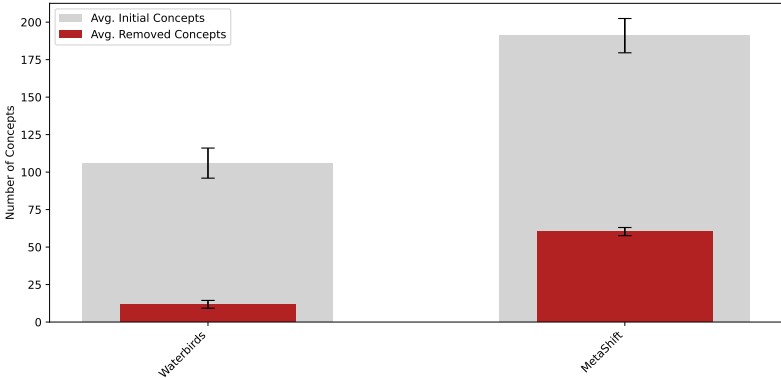

Figure 5: Number of concepts marked as spurious during the debugging sessions for real users on PIP-Net.

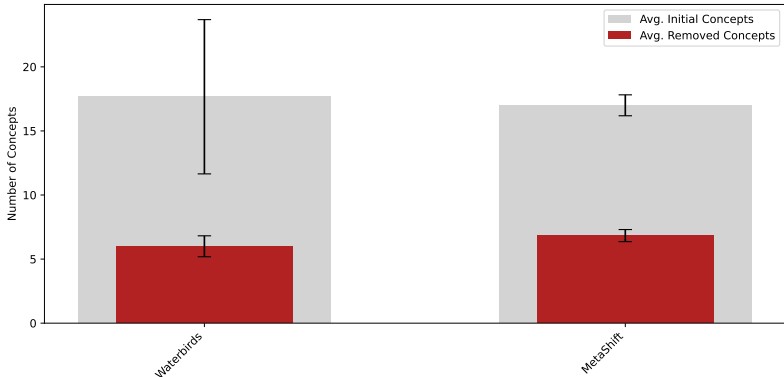

Figure 6: Number of concepts marked as spurious during the debugging sessions for real users on Post-hoc CBM.

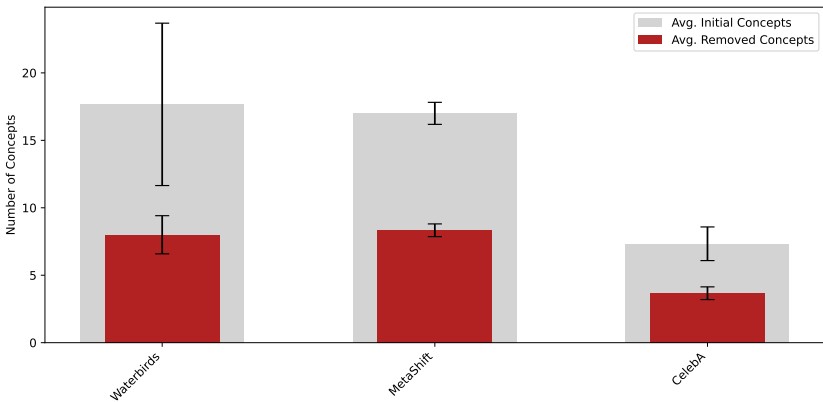

Figure 7: Number of concepts marked as spurious during the debugging sessions for automated users on Post-hoc CBM.

## A.3 RETRAINING

We then describe our retraining process. For PIP-Net we fully fine-tune the original model for an additional 30 epochs. We use a batch size of 64 with a reduced learning rate of $1e^{-5}$ for the backbone, and $0.05$ for the linear layer. For Post-hoc CBM, we follow the original work and keep the backbone and concept layer frozen, only retraining the linear layer. We keep $\lambda_{\text{sparse}} = 0.02$.

We then retrain according to each of the following retraining algorithms:

- **Remove**: Remove the concepts from the concept bottleneck (e.g. zero out weights in linear layer).
- **Retrain**: Perform removal, then continue fine-tuning without performing any intervention for PIP-Net, and retrain linear layer for Post-hoc CBM.
- **ProtoPDebug**: Add a forgetting loss for spurious concept activations directly to the loss term. ProtoPDebug cannot be applied to Post-hoc CBMs because the loss is added to the concept bottleneck layer, but Post-hoc CBM only the final layer is retrained.
- **Augment**: Augment the training dataset and then retrain. For PIP-Net, we randomly select one of the top ten patches for each concept and randomly paste five patches within each image. For Post-hoc CBM, we utilize the synthetic concept bank from Wu et al. (2023) to perform Mixup with, but a text-to-image model could create a concept bank for any text-based concepts used. We augment with a fixed value, keeping $0.75$ of the original image and $0.25$ of the spurious concept image for Mixup.
- **Reweight**: Reweight all samples in the training dataset according to our permutation weighting scheme and then retrain. For PIP-Net these weights are applied to the classification loss, and for Post-hoc CBM these weights are applied when retraining the final linear layer.
- **CBDebug**: Combine both Reweight and Augment and then retrain. Augmentation probabilities are computed by squaring the normalized inverted weights. Reweight all samples according to our permutation weighting scheme, and augment samples that were assigned low weights by our scheme to further reduce the impact of the spurious attribute. We use $\gamma = 2$ for all datasets except ISIC, where we use $\gamma = 5$.

## A.4 DEFINITIONS

Permutation Weighting: A technique that reweights training samples to reduce reliance on a given auxiliary variable. Each sample receives a weight reflecting its likelihood under a permuted version of the auxiliary variable, encouraging the model to focus less on spurious correlations.

Worst-Group Accuracy (WGA): Measures the model's performance on the group of data on which it performs the worst. This metric highlights whether the model fails on minority or atypical subgroups.

# B    FULL CONCEPT BANKS

We begin this section by motivating our methodology for concept bank creation. Prior work (Choi et al.) on the distributional robustness of concept bottlenecks uses synthetic concepts from Wu et al. (2023) as the concept bank for training Post-hoc CBMs. We first note that their subselection of certain concepts requires knowledge of the underlying classification task and so in our work we do not subselect categories, except for ISIC where we only use texture concepts as defined in Wu et al. (2023). Additionally, as can be seen in Figure 4 of Choi et al., there are no concepts that distinguish well between waterbirds and landbirds. Since we are working with real users, a model that learns exclusively spurious concepts where all concepts are removed by the user is not useful.

Below we present the concept bank for each dataset used in this paper. Each dataset utilizes the synthetic concepts from Wu et al. (2023), and then also utilizes our set of curated concepts. We first present the synthetic concept set utilized on all datasets.

---

**Synthetic Concept Set**

*blackness, blueness, greenness, redness, whiteness, concrete, granite, leather, laminate, metal, blotchy, blurriness, stripes, polka dots, knitted, cracked, frilly, waffled, scaly, lacelike, grooved, stratified, gauzy, marbled, flecked, stained, braided, matted, meshed, cobwebbed, spiralled, dotted, crosshatched, wrinkled, woven, potholed, crystalline, paisley, veined, fibrous, studded, bubbly, pleated, grid, perforated, porous, interlaced, smeared, honeycombed, sprinkled, chequered, lined, banded, bumpy, zigzagged, swirly, pitted, freckled, bamboo, beach, bridge, bush, canopy, earth, field, flower, flowerpot, fluorescent, forest, grass, ground, harbor, hill, lake, mountain, muzzle, palm, path, plant, river, sand, sea, snow, tree, water, awning, base, bench, building, earth, fence, field, ground, house, manhole, path, snow, streets, air-conditioner, apron, armchair, back-pillow, balcony, bannister, bathrooms, bathtub, bed, bedclothes, bedrooms, cabinet, carpet, ceiling, chair, chandelier, chest-of-drawers, countertop, curtain, cushion, desk, dining-rooms, door, door-frame, double-door, drawer, drinking-glass, exhaust-hood, figurine, fireplace, floor, flower, flowerpot, fluorescent, ground, handle, handle-bar, headboard, headlight, house, jar, lamp, light, microwave, mirror, ottoman, oven, pillow, plate, refrigerator, sofa, stairs, toilet, bird, cat, cow, dog, horse, mouse, paw, arm, back, body, ear, eye, eyebrow, female-face, leg, male-face, foot, hair, hand, head, inside-arm, knob, mouth, neck, nose, outside-arm, ashcan, airplane, bag, bus, beak, bicycle, blind, board, book, bookcase, bottle, bowl, box, brick, basket, bucket, bumper, can, candlestick, cap, car, cardboard, ceramic, chain-wheel, chimney, clock, coach, coffee-table, column, computer, counter, cup, desk, engine, fabric, fan, faucet, flag, floor, food, foot-board, frame, glass, keyboard, lid, loudspeaker, minibike, motorbike, napkin, pack, painted, painting, pane, paper, pedestal, person, pillar, pipe*

---

To add more useful concepts, we add the attributes from CUB (Wah et al., 2011), translated to natural language, to the concept bank for the Waterbirds dataset. For ISIC, we utilize the eight concepts from Yuksekgonul et al. (2023). For MetaShift and CelebA, we instead curate a set of concepts using a large language model similar to Oikarinen et al. (2023), with a simpler prompt and more powerful model, GPT-4o. We also perform some manual pruning to ensure the concepts are useful for the given task (for example, removing cat and dog from the synthetic concept bank for MetaShift, or removing background concepts from the curated concept bank).

We utilize these concept banks as a proof of concept that CBDebug helps text-based models, and leave further exploration of the impact of different concept banks to future work.

---

**Curated Concept Set Prompt**

You are a concept generation assistant. Generate a list of clear and concise concepts that are important visual features for a 'class_name'. Generate a list of concepts, with each concept appearing on a separate line. Do not include any extra formatting, descriptions, or explanations—just the raw concepts.

Concepts:

---

Then, we show the curated concept set for each dataset.

**Waterbirds Curated Concept Set**

*curved beak, dagger beak, hooked beak, needle beak, hooked seabird beak, spatulate beak, all-purpose beak, cone beak, specialized beak, blue wings, brown wings, iridescent wings, purple wings, rufous wings, grey wings, yellow wings, olive wings, green wings, pink wings, orange wings, black wings, white wings, red wings, buff wings, blue upperparts, brown upperparts, iridescent upperparts, purple upperparts, rufous upperparts, grey upperparts, yellow upperparts, olive upperparts, green upperparts, pink upperparts, orange upperparts, black upperparts, white upperparts, red upperparts, buff upperparts, blue underparts, brown underparts, iridescent underparts, purple underparts, rufous underparts, grey underparts, yellow underparts, olive underparts, green underparts, pink underparts, orange underparts, black underparts, white underparts, red underparts, buff underparts, solid breast, spotted breast, striped breast, multi-colored breast, blue back, brown back, iridescent back, purple back, rufous back, grey back, yellow back, olive back, green back, pink back, orange back, black back, white back, red back, buff back, forked tail tail, rounded tail tail, notched tail tail, fan-shaped tail tail, pointed tail tail, squared tail tail, blue upper tail, brown upper tail, iridescent upper tail, purple upper tail, rufous upper tail, grey upper tail, yellow upper tail, olive upper tail, green upper tail, pink upper tail, orange upper tail, black upper tail, white upper tail, red upper tail, buff upper tail, spotted head, malar head, crested head, masked head, unique pattern head, eyebrow head, eyering head, plain head, eyeline head, striped head, capped head, blue breast, brown breast, iridescent breast, purple breast, rufous breast, grey breast, yellow breast, olive breast, green breast, pink breast, orange breast, black breast, white breast, red breast, buff breast, blue throat, brown throat, iridescent throat, purple throat, rufous throat, grey throat, yellow throat, olive throat, green throat, pink throat, orange throat, black throat, white throat, red throat, buff throat, blue eyes, brown eyes, purple eyes, rufous eyes, grey eyes, yellow eyes, olive eyes, green eyes, pink eyes, orange eyes, black eyes, white eyes, red eyes, buff eyes, about the same as head bill, longer than head bill, shorter than head bill, blue forehead, brown forehead, iridescent forehead, purple forehead, rufous forehead, grey forehead, yellow forehead, olive forehead, green forehead, pink forehead, orange forehead, black forehead, white forehead, red forehead, buff forehead, blue under tail, brown under tail, iridescent under tail, purple under tail, rufous under tail, grey under tail, yellow under tail, olive under tail, green under tail, pink under tail, orange under tail, black under tail, white under tail, red under tail, buff under tail, blue nape, brown nape, iridescent nape, purple nape, rufous nape, grey nape, yellow nape, olive nape, green nape, pink nape, orange nape, black nape, white nape, red nape, buff nape, blue belly, brown belly, iridescent belly, purple belly, rufous belly, grey belly, yellow belly, olive belly, green belly, pink belly, orange belly, black belly, white belly, red belly, buff belly, rounded-wings wings, pointed-wings wings, broad-wings wings, tapered-wings wings, long-wings wings, large size, small size, very large size, medium size, very small size, upright-perching water-like body, chicken-like-marsh body, long-legged-like body, duck-like body, owl-like body, gull-like body, hummingbird-like body, pigeon-like body, tree-clinging-like body, hawk-like body, sandpiper-like body, upland-ground-like body, swallow-like body, perching-like body, solid back, spotted back, striped back, multi-colored back, solid tail, spotted tail, striped tail, multi-colored tail, solid belly, spotted belly, striped belly, multi-colored belly, blue primary color, brown primary color, iridescent primary color, purple primary color, rufous primary color, grey primary color, yellow primary color, olive primary color, green primary color, pink primary color, orange primary color, black primary color, white primary color, red primary color, buff primary color, blue legs, brown legs, iridescent legs, purple legs, rufous legs, grey legs, yellow legs, olive legs, green legs, pink legs, orange legs, black legs, white legs, red legs, buff legs, blue bill, brown bill, iridescent bill, purple bill, rufous bill, grey bill, yellow bill, olive bill, green bill, pink bill, orange bill, black bill, white bill, red bill, buff bill, blue crown, brown crown, iridescent crown, purple crown, rufous crown, grey crown, yellow crown, olive crown, green crown, pink crown, orange crown, black crown, white crown, red crown, buff crown, solid wing, spotted wing, striped wing, multi-colored wing*

**MetaShift Curated Concept Set**

*Long snout, Short snout, Floppy ears, Upright ears, Round eyes, Slit pupils, Curled tail, Straight tail, Stocky body, Slim body, Wide muzzle, Narrow muzzle, Large nose, Small nose, Broad paws, Small paws, Short, dense fur, Fine, soft fur, Simple or spotted coat, Striped or marbled coat, Short whiskers, Long whiskers, Expressive face, Neutral face, Square or upright posture, Crouched or perched posture*

**CelebA Curated Concept Set**

*Light color, Dark color, Low contrast, High contrast, Warm, yellow tones, Cool, brown tones, Light eyebrows, Dark eyebrows, Light lashes, Dark lashes, Finer hair, Thicker hair, Soft texture, Coarse texture, Less visible roots, More visible roots*

**ISIC Curated Concept Set**

*blue-white veil, regular dots and globules, irregular dots and globules, regression structures, irregular streaks, regular streaks, atypical pigment network, typical pigment network*

# C  ABLATIONS

## C.1  PERMUTATION WEIGHTING

Permutation Weighting utilizes two main hyperparameters that we ablate in this section. The first is the number of folds K in K-fold cross-validation. We perform K-fold cross-validation in order to utilize all of our training data to train the classifier while evaluating on unseen data. We also average the weights over multiple random permutations of the dataset to get a more robust estimate of the weights.

We then evaluate the impact of these hyperparameters on the assigned sample weights. In Table 4 and Table 5 we evaluate different combinations of the number of folds and number of permutations, and report the average weight assigned to each subgroup in the Waterbirds dataset. We found that the average weights per group is fairly robust to these hyperparameters, but as you increase the number of folds and decrease the number of permutations, the average weights for the minority subgroups increase. For our main results, we select five permutations and five folds to balance computational cost and robustness.

Table 4: Effect of number of folds (permutations fixed at 5) on average weights for each Waterbirds subgroup for a randomly selected user.

| Class (y)
Background (a) | Landbird
Land | Landbird
Water | Waterbird
Land | Waterbird
Water |
|---|---|---|---|---|
| # Training Samples | 3498 | 184 | 56 | 1057 |
| Average Weight (2 folds) | 1.1 | 2.6 | 5.5 | 0.6 |
| Average Weight (5 folds) | 1.1 | 2.9 | 7.1 | 0.6 |
| Average Weight (10 folds) | 1.1 | 3.2 | 8.3 | 0.7 |

Table 5: Effect of number of permutations (folds fixed at 5) on average weights for each Waterbirds subgroup for a randomly selected user.

| Class (y)
Background (a) | Landbird
Land | Landbird
Water | Waterbird
Land | Waterbird
Water |
|---|---|---|---|---|
| # Training Samples | 3498 | 184 | 56 | 1057 |
| Average Weight (1 permutation) | 1.0 | 2.8 | 7.7 | 0.6 |
| Average Weight (5 permutations) | 1.1 | 2.9 | 7.1 | 0.6 |
| Average Weight (10 permutations) | 1.1 | 2.8 | 6.6 | 0.6 |

## C.2 DOES CBDEBUG EFFECTIVELY REMOVE DEPENDENCE ON SPURIOUS ATTRIBUTES?

We obtain the same visualizations of concepts before and after retraining for all the rest of the datasets and model combinations we test on. We see similar results as before, showing that CBDebug can effectively remove spurious concepts. In Table 6 the original model learns 'bookcase' which correlates highly with being indoors, and the retrained model learns 'bedroom' which correlates highly with being indoors as well. For CBDebug, none of its top five concepts correlate highly with the spurious attribute (indoor vs. outdoor). In Figure 8, we see that the concepts learned by PIP-Net are not as well disentangled on MetaShift compared to Waterbirds (see our discussion in Section A.1). Finally, in Table 7, the original model learns the concept 'male face' for dark hair and 'female face' for blonde hair, but both baseline retraining and CBDebug remove the reliance on these main spurious concepts. CBDebug also learns 'Dark color', which better correlates with dark hair than 'building'.

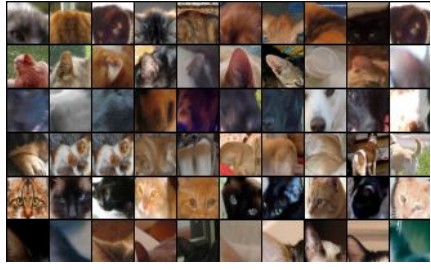
(a) Concepts before retraining

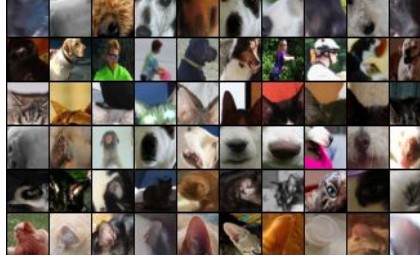
(b) Concepts after retraining with CBDebug

Figure 8: The six most highly activated concepts for the Original model trained on MetaShift and the model after retraining with CBDebug. PIP-Net learns less disentangled concepts on MetaShift, making the intervention less clear visually.

Table 6: Top five concepts for Post-hoc CBM before retraining, after retraining normally, and after retraining with CBDebug on MetaShift.

| Class | Original | Retrain | CBDebug |
|---|---|---|---|
| Cat | Curled tail | Curled tail | Long whiskers |
| | Long whiskers | Long whiskers | Short whiskers |
| | Short whiskers | Short whiskers | Slit pupils |
| | mouse | bedroom | bird |
| | bookcase | Short, dense fur | Curled tail |
| Dog | Floppy ears | Floppy ears | Short snout |
| | Wide muzzle | Wide muzzle | Long snout |
| | Short snout | Short snout | Floppy ears |
| | Narrow muzzle | Narrow muzzle | Wide muzzle |
| | Long snout | Long snout | Narrow muzzle |

Table 7: Top five concepts for Post-hoc CBM before retraining, after retraining normally, and after retraining with CBDebug on CelebA.

| Class | Original | Retrain | CBDebug |
|---|---|---|---|
| Dark Hair | male face | building | Dark color |
| | blackness | counter | granite |
| | box | ceiling | mirror |
| | building | eyebrow | eyebrow |
| | - | pillar | house |
| Blonde Hair | Less visible roots | Less visible roots | Less visible roots |
| | More visible roots | More visible roots | More visible roots |
| | female face | freckled | freckled |
| | - | - | matted |

### C.3 AUGMENTATION PROBABILITIES

To convert the sample weights into probabilities, we first substract each from the max and then normalize them to $[0, 1]$. This ends up with an extremely right-skewed distribution, so we also add a hyperparameter $\gamma$ to control this skew by taking the augmentation probabilities to the power of $\gamma$. We plot the histogram for $\gamma = 1$ and $\gamma = 2$ in Figure 9 with 100 bins on the Waterbirds dataset. Additionally, this hyperparameter enables simple interpolation between our two approaches, because as $\gamma \to \infty$ you do not augment any samples and recover normal permutation weighting.

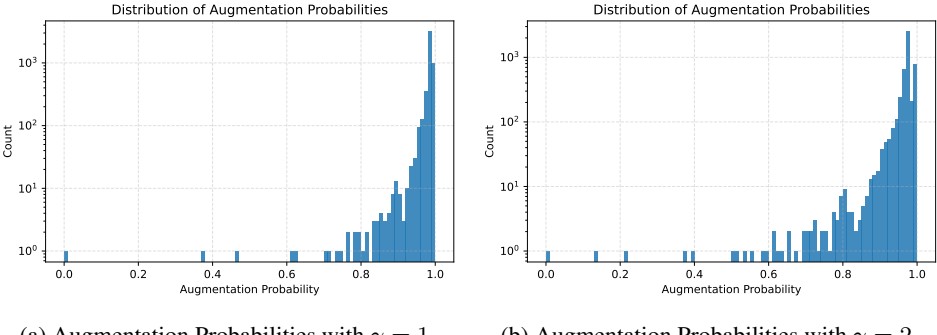

(a) Augmentation Probabilities with $\gamma = 1$.      (b) Augmentation Probabilities with $\gamma = 2$.

Figure 9: Augmentation Probabilities computed on Waterbirds, with counts plotted on a log-scale. Squaring helps reduce the extreme right skew of the probabilities, reducing the probability that non-spurious samples get augmented.

Table 8 shows worst-group accuracy (WGA) on Waterbirds for $\gamma \in \{1, 2, 5, 10\}$. While there is some variability across seeds, WGA remains relatively stable, indicating that CBDebug is robust to the exact choice of $\gamma$. For Waterbirds, MetaShift, and CelebA, we use $\gamma = 2$. For ISIC, we use $\gamma = 5$ to reduce augmentation intensity and avoid distribution shifts caused by applying diffusion-generated concepts to medical images. Extensive tuning of this hyperparameter is left for future work.

Table 8: Worst-Group Accuracy across values of $\gamma$ (augmentation probability) for Post-hoc CBM on Waterbirds.

|  | Average | Worst |
|---|---|---|
| $\gamma = 1$ | 75.3 $_{\pm 6.3}$ | 60.2 $_{\pm 12.7}$ |
| $\gamma = 2$ | 76.0 $_{\pm 2.8}$ | 58.3 $_{\pm 6.0}$ |
| $\gamma = 5$ | 77.6 $_{\pm 6.0}$ | 58.9 $_{\pm 9.1}$ |
| $\gamma = 10$ | 77.8 $_{\pm 6.6}$ | 62.5 $_{\pm 11.4}$ |

## C.4 COMPARISONS TO UNSUPERVISED BIAS MITIGATION APPROACHES

While `CBDebug` offers a distinct approach from popular unsupervised bias mitigation pipelines: giving direct control to a domain expert who interacts with the downstream machine learning model instead of relying on training dynamics to guess what spurious correlations might be present, we do test our approach on bias mitigation datasets, where unsupervised bias mitigation pipelines can serve as a useful benchmark for the effectiveness of `CBDebug`.

We evaluate two unsupervised bias mitigation approaches on Waterbirds and MetaShift with PIP-Net: Just train twice (Liu et al., 2021) (JTT) and Learning from Failure (Nam et al., 2020) (LfF). Following Espinosa Zarlenga et al. (2024), we perform hyperparameter tuning using average validation accuracy, to avoid leaking privileged information about the underlying groups. For JTT, we select the number of epochs $T$ from $(1, 5, 25)$ and the upweighting term $\lambda_{up}$ from $(10, 25, 50)$, and select $(T, \lambda_{up}) = (10, 25)$ for Waterbirds and $(10, 5)$ for MetaShift. For LfF we select the bias amplification term $q$ from $(0.05, 0.1, 0.25, 0.5, 0.75, 0.9, 0.95)$ and select $q = 0.9$ for Waterbirds and $q = 0.95$ for MetaShift.

Our results are shown in Table 9. While JTT shows no meaningful improvement, LfF does improve the worst-group accuracy compared to the original model. `CBDebug` demonstrates a stronger ability to mitigate bias, improving worst-group accuracy over both of these unsupervised pipelines. This highlights `CBDebug`'s effectiveness in leveraging expert feedback on spurious concepts to fine-tune the model.

Table 9: Average and Worst-Group Accuracy on Waterbirds and MetaShift with PIP-Net.

| Method | Waterbirds | | MetaShift | |
|---|---|---|---|---|
| | Average | Worst | Average | Worst |
| Original | $92.3_{\pm 0.3}$ | $71.9_{\pm 2.7}$ | $80.9_{\pm 1.3}$ | $52.4_{\pm 2.0}$ |
| Remove | $92.6_{\pm 0.4}$ | $74.4_{\pm 2.2}$ | $81.4_{\pm 0.6}$ | $55.0_{\pm 2.6}$ |
| Retrain | $92.4_{\pm 0.1}$ | $72.5_{\pm 1.0}$ | $81.2_{\pm 1.6}$ | $53.3_{\pm 2.1}$ |
| ProtoPDebug | $92.5_{\pm 0.1}$ | $71.6_{\pm 1.9}$ | $80.9_{\pm 1.4}$ | $52.4_{\pm 1.4}$ |
| JTT | $91.8_{\pm 0.1}$ | $71.7_{\pm 2.6}$ | $80.7_{\pm 0.5}$ | $51.9_{\pm 1.6}$ |
| LfF | $92.8_{\pm 0.2}$ | $75.4_{\pm 0.8}$ | $81.5_{\pm 0.3}$ | $56.0_{\pm 1.4}$ |
| *Ours* | | | | |
| Reweight Only | $93.2_{\pm 0.4}$ | $74.2_{\pm 4.8}$ | $81.8_{\pm 1.4}$ | $\underline{56.1}_{\pm 1.3}$ |
| Augment Only | $92.4_{\pm 0.6}$ | $\underline{75.5}_{\pm 2.9}$ | $82.2_{\pm 1.7}$ | $55.6_{\pm 3.3}$ |
| **CBDebug** | $93.7_{\pm 0.7}$ | $\mathbf{79.4}_{\pm 4.3}$ | $82.3_{\pm 1.7}$ | $\mathbf{57.3}_{\pm 3.1}$ |

