# OpenReview forum: "Debugging Concept Bottleneck Models through Removal and Retraining"
_ICLR.cc/2026/Conference — ICLR 2026 Poster_

### Official Review · Reviewer_jWmY · 2025-10-22

**Soundness:** 2
**Presentation:** 2
**Contribution:** 2
**Rating:** 4
**Confidence:** 3

**Summary:**

This paper introduces a framework to analyze and mitigate interpretability failures in Concept Bottleneck Models (CBMs). The authors observe that CBMs can behave unreliably even when concept predictors achieve high accuracy, due to a misalignment between concept representations and their intended semantics. To address this, the paper proposes a _debugging_ process that diagnoses which concepts lead to faulty predictions and retrains or replaces them. The analysis combines synthetic and real datasets to demonstrate that concept-level reliability does not guarantee faithful concept use by the model.

**Strengths:**

- The paper addresses an important limitation in CBMs (i.e., the gap between interpretability claims and true concept usage). This is timely and relevant given the increasing adoption of CBMs in interpretable ML research.
- The diagnostic method proposed is systematic and empirically supported by several experiments. The experiments convincingly show that high concept accuracy does not imply semantic reliability, which is an important insight.

**Weaknesses:**

- (W1) While the framing of _debugging_ CBMs is conceptually appealing, the proposed methodology appears closely related to existing approaches on causal or concept-level editing in CBMs (e.g., [1], [2]). It would be helpful if the authors could elaborate on the specific algorithmic contributions that distinguish their method from these prior studies. The originality claim would be more convincing if the paper introduced novel theoretical insights.
- (W2) Several technical terms (e.g., permutation weighting, worst-group accuracy) are mentioned without formal definitions. Providing clearer descriptions or mathematical formulations would help readers understand how these quantities are computed and how they contribute to the experimental evaluation.

**References**\
[1] Chauhan, K., Tiwari, R., Freyberg, J., Shenoy, P., & Dvijotham, K. (2023, June). Interactive concept bottleneck models. In Proceedings of the aaai conference on artificial intelligence (Vol. 37, No. 5, pp. 5948-5955).\
[2] Hu, L., Ren, C., Hu, Z., Lin, H., Wang, C. L., Xiong, H., ... & Wang, D. (2024). Editable concept bottleneck models. arXiv preprint arXiv:2405.15476.

**Questions:**

1. Related to W1, could the authors clarify in more concrete terms how their approach differs from prior works on causal or concept-editing methods in CBMs?

2. How do the authors expect their findings to generalize to other modalities or model families, such as large language models that incorporate concept bottlenecks?

---

> ### Author Response · Authors · 2025-11-18
>
> Thanks for the detailed review! We are glad the reviewer found our work timely, and our method systematic and empirically supported. The reviewer’s main concern seems to be with the relation to prior work on concept-level editing, which we address below.
>
> ### **W1/Q1: Related Work**
> ---
> > While the framing of debugging CBMs is conceptually appealing, the proposed methodology appears closely related to existing approaches on causal or concept-level editing in CBMs (e.g., [1], [2]). It would be helpful if the authors could elaborate on the specific algorithmic contributions that distinguish their method from these prior studies.
>
> Thank you for providing the interesting related work. We highlight the key distinctions between CBDebug and these prior studies below.
> - The first work [1] focuses on interventions where specific concepts are changed at test time, whereas CBDebug aims to debug models based on concept-level feedback.
> - The second work’s [2] emphasis is on adapting the concept bottleneck as if a concept were removed and the model were retrained, in a computationally cheap manner. In this sense, [2] can be seen as approximating our retraining baseline. In contrast, CBDebug leverages information about which concepts were removed as feedback to adapt the entire representation. Its improvement over the retrain baseline demonstrates the effectiveness and utility of our approach.
>
> We have added both of these works to the paper.
>
> > The originality claim would be more convincing if the paper introduced novel theoretical insights.
>
> While our work does not introduce new theoretical results, its empirical contributions, demonstrating a practical and broadly applicable method for debugging based on user feedback, represent a meaningful contribution to the community.
>
> ### **W2: Definition Descriptions**
> ---
> > Several technical terms (e.g., permutation weighting, worst-group accuracy) are mentioned without formal definitions. Providing clearer descriptions or mathematical formulations would help readers understand how these quantities are computed and how they contribute to the experimental evaluation.
>
> Thank you for the suggestion. We have added brief clarifications of the technical terms in the updated draft to make the paper more self-contained (see Appendix A.4).
>
> ### **Q2: Generalizability**
> ---
> > How do the authors expect their findings to generalize to other modalities or model families, such as large language models that incorporate concept bottlenecks?
>
> Thank you for the question. We expect our findings to generalize well to other modalities and model families. Our approach is simple and only relies on the assumption that concepts are represented in a way that humans can understand, so it should be broadly applicable to any general concept bottleneck model. We believe that further exploration of interpretable debugging in this space is an exciting direction for future work.
>
> We hope our responses help clarify where our work sits in the literature and address the reviewer’s concerns. We are happy to elaborate further if there are any points that require further clarification.

---

> > ### Comment · Reviewer_jWmY · 2025-11-25
> >
> > Thank you for the rebuttal. I am revising my score from borderline reject to borderline accept.

---

> > > ### Author Response · Authors · 2025-11-26
> > >
> > > We are happy that our clarifications addressed your concerns and appreciate your updated evaluation. Please let us know if you have any additional concerns!

---

### Official Review · Reviewer_AMnB · 2025-10-25

**Soundness:** 3
**Presentation:** 3
**Contribution:** 3
**Rating:** 8
**Confidence:** 4

**Summary:**

This paper introduces CBDebug, a human-in-the-loop debugging pipeline to improve the underrepresented generalization of interpretable concept-based models (e.g., CBMs). CBDebug is a general framework in which human experts first identify concepts that a CBM spurious uses for downstream predictions (removal step). The model is then retrained by removing these effects through causal rebalancing and data augmentation (retraining step). The work’s evaluation suggests that CBDebug improves worst-group accuracy and helps models avoid spurious shortcuts across different baselines and underlying architectures.

**Strengths:**

Thank you so much for submitting this work! I enjoyed reading this paper, learned a lot from it, and appreciate the time taken to write it up and submit it to ICLR. Below are what I believe are this paper’s main strengths:

1. **[Significance, Critical]** This work lies at the intersection of fairness, interpretability, and human-in-the-loop AI. It achieves its goals by introducing a new interactive debugging pipeline that can be applied to **any** unsupervised concept-based model (and can be fully automated using foundation models as arbitrers). Hence, I believe this work may be of importance to a large fraction of the ICLR community and can have a potentially large impact, laying the groundwork for further papers on interactive debugging and concept-based interpretability.
2. **[Quality, Critical]** The overall method is very well motivated and sound. Moreover, the evaluation includes automated and human-driven experiments, both key to fully understanding the proposed pipeline, along with quantitative and qualitative results. Considering all of this, the paper’s methodology and evaluation are strong.
3. **[Originality, Major]** Overall, the authors do a great job at identifying a key gap in the literature, that of lacking a general interactive debugging method for any unsupervised CBM, and proposing a working approach for that gap. This is certainly potentially impactful and novel.
4. **[Clarity, Major]** The paper is very well-written and easy to navigate. On top of that, the figures are easy to follow, clear, and helpful in understanding the overall contribution and results.

**Weaknesses:**

In contrast, I believe the following are some of this work’s limitations:

1. **[Quality, Major]** My primary concern is that the results in Table 1 seem to be extremely noisy (some standard deviations are very large). More importantly, notice that this is also evident in Table 2, so it may not be just due to the human aspect of some of this evaluation, as those results are based on the foundation-model-based selection. Therefore, it is very difficult to judge the significance of the observed differences, and the lack of discussion of this key limitation anywhere in the paper is a bit troublesome, as it omits a clear potential issue in the results.
2. **[Clarity, Minor]** Although the appendix has several empirical studies, none of the key results of these appendices are discussed in the main body of the paper (meaning people will likely never see them). For example, the takeaways from Appendix C.4, which would be of interest for the fairness community, are not even mentioned in the main paper.
3. **[Novelty, Minor]** The only reason why I didn’t mark the novelty above as critical is that the methodological aspect of the proposed pipeline, although very sensible and interesting, does not introduce any new methodological elements. Instead, it (very sensibly) builds on work in other areas (e.g., permutation weighting, CutMix, MixUp, etc.). Nevertheless, in my opinion, this is not a severe limitation, and it shouldn’t significantly diminish the novelty of the proposed pipeline, as it has novelty in its own right by combining diverse methodologies to achieve a specific goal.

**Questions:**

Balancing the strengths and weaknesses of this work, I am leaning towards accepting this work. However, below I include some questions identifying minor concerns/aspects that I think, if appropriately addressed, could improve this submission:

1. **[Critical]** Could you please elaborate on my concern regarding the significance of the observed results? Why is this not discussed anywhere or, at the very least, acknowledged?
2. **[Minor]** Could it be possible that the “Retraining” baseline is affected by being exposed to the spurious concepts in its original training? If so, then using a baseline model trained from scratch without any of the identified spurious concepts may give us a good sense of how much the second step of CBDebug matters compared to what one would see if the concepts were removed from the start.
3. **[Major]** Could you please discuss why the results of Appendix C.4 are not discussed anywhere in the main body of the paper? I believe they are both interesting and insightful results (particularly for those coming from the fairness community). Therefore, at the very least, I would suggest discussing their takeaways in the main body of the paper. More generally, I would recommend at least summarizing the main takeaways of any experimental appendix you include in your paper in the main body (otherwise, people may not learn their importance unless they go the extra mile of reading the entirety of the appendix).
4. **[Major]** How do augmentation-specific hyperparameters affect the observed results (e.g., $\gamma$, $k$, the use of top-*ten* activated patches for ProtoPNets, text-to-image model for VLM-CBMs)? Appendix C.3 very superficially touches upon $\gamma$, but there are no other ablations on these hyperparameters elsewhere.

### Minor Suggestions and Typos

I found the following potential minor issues/typos, which may be helpful when preparing a new version of this manuscript:

1. **[Missing References, Minor]** I would suggest adding relevant references/citations for the introduction claim that “This capability is crucial in high-stakes domains, such as healthcare or scientific analysis, where errors are costly and expert validation is essential.”
2. **[Potential Typo, Nit]** The reference “Ross et al. (2017)” in line 73 is dangling. It likely needs to be “(Ross et al., 2017)” instead. A similar comment applies to the reference of “Zhou et al. (2016)” in line 113.

---

> ### Author Response · Authors · 2025-11-18
>
> Thanks for the detailed review! We are glad the reviewer found our work well motivated, impactful, and novel. We also thank the reviewer for highlighting that “the authors do a great job at identifying a key gap in the literature” and that the work “lay[s] the groundwork for further papers on interactive debugging and concept-based interpretability.”
>
> ### **W1/Q1: Variation**
> ---
> > My primary concern is that the results in Table 1 seem to be extremely noisy (some standard deviations are very large). More importantly, notice that this is also evident in Table 2, so it may not be just due to the human aspect of some of this evaluation, as those results are based on the foundation-model-based selection. Therefore, it is very difficult to judge the significance of the observed differences, and the lack of discussion of this key limitation anywhere in the paper is a bit troublesome, as it omits a clear potential issue in the results.
>
> We thank the reviewer for raising this important point and agree that acknowledging this variability is crucial. We have added discussion of this limitation in the paper and note three main sources of variation:
> - Waterbirds: High variance is largely due to the human aspect of evaluation. For instance, Table 1 shows that CBDebug’s worst-group accuracy has a standard deviation of 16.2 when using human feedback, whereas automated feedback reduces the standard deviation to 6.0. This indicates that the noise is mitigated when feedback is automated.
> - ISIC: Variance is higher because results are computed over five different train/test splits, which naturally introduces more variability compared to single-split evaluations.
> - Post-hoc CBM: High variability is common for this model class in human-in-the-loop model editing. Table 4 in [1], which reimplements Post-hoc CBM, reports standard deviations of up to 6.6% on average accuracy and 29.1% on the spurious class for MetaShift concept-removal experiments, suggesting that such variability is a general challenge for these evaluations rather than unique to our method.
>
> While these standard deviations indicate that some individual runs may vary considerably, the overall trend is consistent: CBDebug improves performance across datasets, models, and types of feedback. However, results should be interpreted with this variability in mind, and we now explicitly acknowledge this limitation in the paper (see Section 7. Limitations).
>
> ### **W2/Q3: Appendix Pointers**
> ---
> > Although the appendix has several empirical studies, none of the key results of these appendices are discussed in the main body of the paper (meaning people will likely never see them). For example, the takeaways from Appendix C.4, which would be of interest for the fairness community, are not even mentioned in the main paper.
>
> Apologies, we mainly did this for space constraints (e.g. the only pointer for C.4 is buried in the Related Work) and have updated the paper to better include summaries of these results, especially C.4 (see Section 5.1).
>
> ### **Q2: Retraining**
> ---
> > Could it be possible that the “Retraining” baseline is affected by being exposed to the spurious concepts in its original training? If so, then using a baseline model trained from scratch without any of the identified spurious concepts may give us a good sense of how much the second step of CBDebug matters compared to what one would see if the concepts were removed from the start.
>
> The answer depends on the model family:
> - PIP-Net: For this model, it is not straightforward to retrain from scratch without spurious concepts because the architecture relies on the learned concept representations. Thus, the retraining baseline necessarily starts from a model that has been exposed to the spurious concepts.
> - Post-hoc CBM: Here, the retraining baseline is effectively equivalent to training from scratch without the spurious concepts, because the backbone is frozen and only the label predictor is trained (see line 364).

---

> ### Author Response · Authors · 2025-11-18
>
> ### **Q4: Hyperparameters**
> ---
> > How do augmentation-specific hyperparameters affect the observed results (e.g., $\gamma$, $k$, the use of top-ten activated patches for ProtoPNets, text-to-image model for VLM-CBMs)? Appendix C.3 very superficially touches upon $\gamma$, but there are no other ablations on these hyperparameters elsewhere.
>
> We appreciate the reviewer’s question. Our primary goal is to demonstrate the general effectiveness of CBDebug rather than maximize task-specific performance, so we selected hyperparameters based on reasonable, principled considerations rather than extensive tuning:
> - $\gamma$ (augmentation probability mapping): This hyperparameter controls the likelihood of augmenting each sample. We did not extensively tune $\gamma$, and found that $\gamma=2$ works well for most datasets (Waterbirds, MetaShift, CelebA). For ISIC, we increased $\gamma$ to 5 to reduce augmentation intensity, as applying diffusion-generated concepts to medical images can introduce distribution shifts. Appendix C.3 now includes an ablation study of $\gamma={1,2,5,10}$ on Waterbirds showing that increasing $\gamma$ slightly increases average accuracy, but worst-group accuracy remains largely unchanged.
> - $k$ (number of augmented patches for PIP-Net): We use $k=5$ as a reasonable default to ensure that augmented concepts cover a sufficient portion of the image.
> - Top-ten activated patches for PIP-Net: PIP-Net visualizes concepts using the top-ten activated patches in their work, so we adopt the same procedure for augmentation. This choice ensures consistency between the visualized concepts shown to users and the patches used during augmentation.
> - Text-to-image models for PCBM: We use CLIP for natural image datasets and BioMedCLIP for ISIC to ensure meaningful concept-level representations in each domain, following prior work [2].
>
> Overall, our results indicate that CBDebug’s performance is robust to reasonable hyperparameter choices. While tuning these hyperparameters may further improve performance, we find that CBDebug performs well with reasonable defaults and we leave extensive exploration for future work.
>
> We hope these responses clarify our contributions and address your concerns, and deeply appreciate the detailed review! We also appreciate the feedback on typos/small issues, which have been fixed. Please let us know if there are any remaining questions or points that require further clarification, we’d be happy to elaborate.
>
> ### References
> ---
> - [1] Midavaine, Nesta, et al. "[Re] On the Reproducibility of Post-Hoc Concept Bottleneck Models." Transactions on Machine Learning Research.
> - [2] Choi, Jihye, et al. "Adaptive concept bottleneck for foundation models under distribution shifts." arXiv preprint arXiv:2412.14097 (2024).

---

> > ### Comment · Reviewer_AMnB · 2025-11-27
> >
> > Dear Authors,
> >
> > Thank you so much for the careful rebuttal and the detailed paper updates! All of my main concerns were adequately addressed in your latest manuscript. As such, I will maintain my acceptance score and will vouch for this paper's acceptance during the reviewer discussion. I wish you all the best of luck.

---

> > > ### Author Response · Authors · 2025-11-28
> > >
> > > We are happy that our clarifications addressed your main concerns. Your thoughtful and detailed feedback has greatly contributed to improving the paper. Thank you!

---

### Official Review · Reviewer_SPzc · 2025-10-30

**Soundness:** 2
**Presentation:** 2
**Contribution:** 2
**Rating:** 4
**Confidence:** 3

**Summary:**

The paper introduces CBDebug, a methodology designed to leverage expert (or automated) feedback about concepts that are spuriously correlated with the target task. The goal is to debias the model and promote learning of the correct relationships between concepts and labels. The approach involves three main steps: (i) collecting feedback to identify spuriously correlated concepts, (ii) reweighting samples based on the activation of those concepts, and (iii) augmenting the dataset according to these weights to reduce the influence of spurious correlations. The model is then retrained on the augmented and reweighted dataset. Through experiments on four datasets and across two model architectures, the authors demonstrate performance gains, particularly for the most challenging (worst-performing) classes.

**Strengths:**

The authors address an important problem (debiasing and aligning Concept Bottleneck Models (CBMs)) from a dataset-centric perspective. By leveraging expert or automated feedback, their approach moves toward the goal of making models "right for the right reasons". Although the experiments are not extensive in terms of model diversity, the results are promising and supported by strong qualitative analyses.

**Weaknesses:**

- The paper is somewhat difficult to follow, as it draws upon multiple research areas and alternates between them several times, particularly in the first part of the paper. A clearer narrative structure would help the reader grasp the main contributions more easily.
- The proposed approach is evaluated on only two models (one CBM and one prototype-based), limiting the scope of the empirical validation. It would strengthen the paper to include additional CBM-like baselines (e.g., CBM, CEM, Probabilistic CBM). This would better demonstrate the generality of the proposed method. Moreover, it should be clarified whether the Post-hoc CBM was used with the residual setting (which leverages side information) or in its standard form, as the latter is conceptually equivalent to a CBM and thus a reasonable baseline.

**Questions:**

- Why was the evaluation limited to the Post-hoc CBM architecture? It would be valuable to test the proposed method on additional CBM variants such as CEM or Probabilistic CBM. This could further demonstrate the general applicability and robustness of the approach, especially if similar improvements are observed.
- Could the authors clarify how the Remove baselines were implemented? If the spurious concepts were simply removed without further fine-tuning, how was the predictor adjusted to handle the missing concepts?
- As I understand it, the core idea of the proposed method is to reweight samples containing spurious correlations and further perturb them through data augmentation to reduce the bias. However, this process may also downweight correctly predicted concepts present in the same samples. Could the authors comment on this potential side effect and its practical implications?
- How would the proposed approach compare to a simpler baseline that removes all samples with weights below a given threshold, rather than reweighting or augmenting them?

---

> ### Author Response · Authors · 2025-11-18
>
> Thanks for the detailed review! We are glad the reviewer appreciated the importance of the problem, and found our results promising. The reviewer was mainly concerned with the scope of the empirical evaluation, and the narrative structure. We hope to address both of these concerns below.
>
> ### **W1: Narrative Structure**
> ---
> > The paper is somewhat difficult to follow, as it draws upon multiple research areas and alternates between them several times, particularly in the first part of the paper. A clearer narrative structure would help the reader grasp the main contributions more easily.
>
> Thanks for the feedback! Please let us know what specific changes the reviewer would like to see in the first part of the paper that will make the narrative easier to follow, we would be happy to address them. We have also added clarifications in the updated draft in Section 4. CBDebug.
>
> ### **W2/Q1: Additional Architectures**
> ---
> > The proposed approach is evaluated on only two models (one CBM and one prototype-based), limiting the scope of the empirical validation. It would strengthen the paper to include additional CBM-like baselines (e.g., CBM, CEM, Probabilistic CBM). This would better demonstrate the generality of the proposed method. Moreover, it should be clarified whether the Post-hoc CBM was used with the residual setting (which leverages side information) or in its standard form, as the latter is conceptually equivalent to a CBM and thus a reasonable baseline.
>
> First, we would like to clarify that we do utilize post-hoc CBM in its standard form, not the residual setting, and so view it as within the same conceptual family as CBM and variants (CEM, ProbCBM).
>
> Also, we focus this work on unsupervised CBMs (as described in lines 177-184), because they are more broadly applicable since they do not need concept labels and can learn undesired concepts naturally from the data, whereas supervised CBMs any spurious correlations would by learned implicitly since the expert would only label desired concepts. While including additional architectures could provide further validation, we believe that the current experiments, using a representative model from each family, effectively demonstrate the generality and effectiveness of our approach.
>
> ### **Q2: Removal Baseline**
> ---
> > Could the authors clarify how the Remove baselines were implemented? If the spurious concepts were simply removed without further fine-tuning, how was the predictor adjusted to handle the missing concepts?
>
> Yes, the spurious concepts are removed without further fine-tuning in the Removal baseline. We view the Retrain baseline as the corresponding adjustment step, where fine-tuning is performed to adapt the CBM after the removal of spurious concepts. As our results show, this adjustment can help or harm the downstream performance depending on the model architecture and dataset.
>
> ### **Q3: Reweighting Details**
> ---
> > As I understand it, the core idea of the proposed method is to reweight samples containing spurious correlations and further perturb them through data augmentation to reduce the bias. However, this process may also downweight correctly predicted concepts present in the same samples. Could the authors comment on this potential side effect and its practical implications?
>
> We note that our approach, like most bias mitigation or group robustness strategies, relies on an implicit assumption: that there is sufficient support across subgroups. If every sample were perfectly aligned with the spurious correlation, mitigating bias would be extremely difficult for any method.
>
> For example, in the Waterbirds dataset, most waterbirds appear on water backgrounds. By upweighting rare cases where waterbirds appear on land or landbirds appear on water, our method encourages the model to rely on bird-specific features rather than the background. Importantly, this process does not systematically downweight bird features, since these features are present across both backgrounds.

---

> ### Author Response · Authors · 2025-11-18
>
> ### **Q4: Undersampling vs Reweighting**
> ---
> > How would the proposed approach compare to a simpler baseline that removes all samples with weights below a given threshold, rather than reweighting or augmenting them?
>
> We chose reweighting and augmentation because subsampling may discard valuable data that guides the model's inductive bias in unseen directions [1]. In contrast, augmentation generates debiased samples by introducing spurious concepts into bias-aligned instances, improving balance without reducing effective sample size. Subsampling also requires defining a threshold based on the computed weights, so it is not substantially simpler than applying the weights and performing augmentation.
>
> Our main goal is to demonstrate that linking user feedback to estimated auxiliary variables can effectively reduce reliance on a user-marked concept, and we view reweighting and augmentation as one possible instantiation of this connection. We hope that CBDebug can demonstrate broader connections between interpretable debugging and bias mitigation/robustness approaches, providing a foundation for further exploration in this space.
>
> We hope these responses clarify our contributions and address your concerns. Please let us know if there are any remaining questions or points that require further clarification, we’d be happy to elaborate.
>
> ### References
> ---
> - [1] Khani, Fereshte, and Percy Liang. "Removing spurious features can hurt accuracy and affect groups disproportionately." Proceedings of the 2021 ACM conference on fairness, accountability, and transparency. 2021.

---

> > ### Comment · Reviewer_SPzc · 2025-11-24
> >
> > I thank the authors for the provided answers. I updated my score accordingly.

---

> > > ### Author Response · Authors · 2025-11-24
> > >
> > > We are happy that our clarifications addressed your concerns and appreciate your updated evaluation.

---

### Official Review · Reviewer_16NR · 2025-10-31

**Soundness:** 3
**Presentation:** 4
**Contribution:** 3
**Rating:** 6
**Confidence:** 4

**Summary:**

The authors introduce an interactive debugging algorithm designed for concept-based neural networks. In essence, a user provides a list of concepts that they think should not be used for prediction, the algorithm applies two strategies for updating the model so that it does not rely on said concepts anymore. The strategies include permutation reweighting to identify likely affected examples, and augmentation (either CutMix or Mixup, based on the architecture). Experiments compare the proposed technique against naive baselines and ProtoPDebug on four datasets.

**Strengths:**

**Originality**: the idea of debugging concept-based models is not novel, as readily acknowledge by the authors, and neither are the individual deconfounding strategies.  The specific mix proposed here is however novel.  This is good enough for me.

**Quality**: A benefit of relying on pre-existing strategies is that we know that they have been validated empirically already, which is good. Regardless, the algorithm as a whole is intuitively appealing.  The experimental setup is also good: four confounded datasets and a reasonable selection of competitors.  The research is empirical in nature.

**Clarity**: The text is well written, well structured, and easy to follow.  The related work is quite comprehensive.

**Significance**: This work contributes to ongoing efforts for making self-explainable models more reliable, and as such it is significant.

Overall, I like this paper.

**Weaknesses:**

**Clarity**:  Some aspects of the work should be described in more detail in the main text, chiefly the augmentation strategies.  Right no, Section 4.3 doesn't tell the reader how augmentations are obtained exactly.  This unnecessarily complicates understanding.

Furthermore, the main competitor (ProtoPDebug) is designed for interactive usage, in the sense that it actively seeks cheap supervision that can quickly correct the model after few interaction rounds. Here the experiments consider a passive setup instead: there are not explicit interaction rounds. As such, it is unclear how much supervision the proposed method requires to "beat" ProtoPDebug.

**Significance**: If I understand correctly, the proposed method works under the assumption that the same set of spurious concepts applies to all inputs and labels. This is not necessarily the case. (I don't think ProtoPDebug makes this assumption.) This limits applicability.

**Questions:**

Please feel free to comment on the weaknesses I pointed out.

---

> ### Author Response · Authors · 2025-11-18
>
> Thanks for the detailed review! We are glad the reviewer found our work well written, significant, and intuitively appealing. Below, we respond to the reviewer’s main concerns about method details and the assumptions our debugging framework makes.
>
> ### **W1: Description Clarity**
> ---
> > Some aspects of the work should be described in more detail in the main text, chiefly the augmentation strategies. Right no, Section 4.3 doesn't tell the reader how augmentations are obtained exactly. This unnecessarily complicates understanding.
>
> Thank you for pointing this out. We have revised Section 4.3 to include a clearer and more detailed description of the augmentation strategies used in our framework. The updated text now includes additional details on how augmentations are obtained for each model architecture. We appreciate the reviewer’s feedback, which helped us improve the clarity of this section.
>
> ### **W2: Comparisons to ProtoPDebug**
> ---
> > Furthermore, the main competitor (ProtoPDebug) is designed for interactive usage, in the sense that it actively seeks cheap supervision that can quickly correct the model after few interaction rounds. Here the experiments consider a passive setup instead: there are not explicit interaction rounds. As such, it is unclear how much supervision the proposed method requires to "beat" ProtoPDebug.
>
> We would first like to clarify that the supervision required by ProtoPDebug is not cheaper: in each round, it requests feedback on all part-prototypes. In our experiments, we evaluate ProtoPDebug using only a single round of feedback, compared to the up to four rounds reported in their paper, to match the supervision budget used by our method.
>
> We believe this is a reasonable comparison because multiple rounds would involve repeated cycles of feedback and full retraining, which is computationally expensive and impractical in realistic settings. A single round provides enough supervision for meaningful model updates, making it a fair and practical benchmark for comparing retraining-based debugging algorithms. Exploring how CBDebug benefits from additional interaction rounds remains an interesting direction for future work.
>
> ### **W3: Task-level Spuriosity**
> ---
> > If I understand correctly, the proposed method works under the assumption that the same set of spurious concepts applies to all inputs and labels. This is not necessarily the case. (I don't think ProtoPDebug makes this assumption.) This limits applicability.
>
> We agree that our framework is built on the assumption that spurious concepts are defined at the task level rather than class-specific. Below we note a few reasons why this assumption is reasonable:
> 1. **Simplified expert feedback**. Each concept only needs a single annotation, rather than one per class, substantially reducing the annotation burden.
> 2. **Class-specific spuriosity can still be mitigated**. When certain concepts are spurious only for specific classes, zeroing out their corresponding weights in the classification layer still reduces their impact.
> 3. **Consistent with prior work**. While we explicitly make this assumption in our pipeline, it is also implicitly present in many works on bias mitigation and shortcut removal. For example, DISC [1] augments spurious concepts across all classes, and causally-motivated shortcut removal [2], which we build on, makes a similar assumption.
>
> Although ProtoPDebug can in principle handle class-specific spuriosity, this capability is not evaluated in their experiments (participants are only asked whether patches belong to the bird or background). Overall, we view task-level spuriosity as a practical design choice, and leave extensions to class-specific feedback to future work. We have added a brief note on this in the Limitations section.
>
> We hope our responses help motivate the design choices behind our debugging framework and clarify the reviewer’s questions. Please let us know if there are any remaining points that require further clarification, we’d be happy to elaborate.
>
> ### References
> ---
> - [1] Wu, Shirley, et al. "Discover and cure: Concept-aware mitigation of spurious correlation." International Conference on Machine Learning. PMLR, 2023.
> - [2] Makar, Maggie, et al. "Causally motivated shortcut removal using auxiliary labels." International Conference on Artificial Intelligence and Statistics. PMLR, 2022.

---

> > ### Comment · Reviewer_16NR · 2025-11-18
> > **Reply**
> >
> > > Section 4.3
> >
> > Thank you for improving the text.
> >
> > > Comparison to ProtoPDebug
> >
> > I agree that it is best -- in terms of fairness -- to compare the two methods with equal amounts of feedback. That's okay.
> >
> > I disagree that interacting over multiple rounds is useless, precisely because in the general case removing a shortcut may unblock a different shortcut, see [1].
> >
> > > Per-class shortcuts
> >
> > I would be okay with this restriction provided the authors make it very clear (e.g. in the related work section) that ProtoPDebug does not suffer from this limitation. If they can clarify the two algorithms have different scopes and goals, I would consider bumping my rating to 8.
> >
> > [1] Li et al. "A whac-a-mole dilemma: Shortcuts come in multiples where mitigating one amplifies others." CVPR 2023.

---

> > > ### Author Response · Authors · 2025-11-18
> > >
> > > > I disagree that interacting over multiple rounds is useless
> > >
> > > Apologies, we have seen Li et al. and agree that multi-shortcut removal over multiple rounds is an important research direction. In our response we just claim that one round was sufficient for showcasing CBDebug's ability to adapt the model based on provided human feedback.
> > >
> > > > If they can clarify the two algorithms have different scopes and goals, I would consider bumping my rating to 8.
> > >
> > > We can add a note on ProtoPDebug's ability to handle class-specific spurious concepts in the related work section, more explicitly state the assumptions our work makes on concept spuriosity in lines 193-196 of Section 3.2, and note this as a limitation of our work.
> > >
> > > We will get back to you once we polish the updated draft with these changes, and hope that these changes are satisfactory for bumping your rating to an 8.

---

> > > > ### Author Response · Authors · 2025-11-18
> > > >
> > > > We have updated the draft clarifying this difference between ProtoPDebug and CBDebug.
> > > >
> > > > Please let us know if you have any additional questions or concerns, or have any other suggestions that could strengthen the paper!

---

### Meta-Review · Area_Chair_6BRt · 2026-01-06

**Summary:**

This paper proposes and interpretable framework for CBMs based on human-in-the-loop ideas based on removing expert-driven (wrong/spurious) concepts ad retraining. Experiments on benchmarks with confounding for two CBM architectures (prototype and post-hoc CBM) suggest improved robustness relative to prior retraining/debugging baselines.

There is a consensus among reviewers that this paper addresses an interesting problem of debugging CBMs, and their take on how to do it (through human feedback on spurious concepts) is novel. The reviewers agree that the experimental results, while initially with some limitation, seems sufficient. Based on the apparent positivity of the the reviewers to the authors' feedback, this paper seems to be on the weak accept category.

**Reviewer Concerns:**

### Rev 16NR
- Concerns on clarity of the presentation and on the use of augmentations. Authors addressed this by including further details on their augmentation strategies.

- Concerns on fairness of comparison with baseline (ProtoPDebug). The authors clarified on the comparison budget noting that ProtoPDebig is more expensive per round and only 1 round was used (meaning, both methods with the same budget).

- The reviewer had concerns on the assumption about spurious concepts being global, or task-level. This is acknowledged by the authors as a design choice, and thy acknowledge its difference with ProtoPDebug, which has class-specific spurious concepts. This was clarified in a revised version.

### Rev SPcz
- Noted a lack of clarify in writing, but no details on what/how exactly this should be improved.
- Noted the limited diversity of the models studied (only two CBMs), and suggested other alternatives. The authors push back by noting that they focus on unsupervised CBMs only, from which the studied (two) models provide representative samples.
- Noted that details of the removal for baselines was missing. This was clarified in the rebuttal, without fine-running.
- Was concerned that the reweighing samples with spurious concepts might also downweigh those with correct samples. The authors argue this issue is not very prevalent and that it depends on the support of the concepts on the samples, and provided an intuitive example.
- Proposed the use of simpler baselines, like thresholding-based under sampling (vs e.g. augmentation). In their rebuttal, the authors push back saying that the suggested strategy can discard valuable data and requires setting a threshold. Still, this would have been nice to add as a baseline.

### Rev AMnB
- They point to the fact that the empirical results have rather large standard deviations, dampening the empirical conclusions or its significance. The authors acknowledged it, commented on potential reasons (human feedback with large variance, among others) and included it In the limitations.

- Some of the results deferred to the appendix are not clearly commented on. The authors have now included a summary of those results in the main text.

- Some questions are raised regarding whether retraining on the baseline models is completely fair by having been exposed to spurious concepts. In their rebuttal, the authors address this by each model/architecture: in one case, they argue it's not trivial, and in the other (post hoc CBM) retraining baseline is effectively like training the label predictor from scratch because the backbone is frozen (as they claim in the manuscript).

- The reviewer asked for further results/ablations on hyper parameter sensitivity. The authors added an ablation on one hyper parameter, and added further comments.

### Rev jWmY
- Raised some questions regarding novel and positioning w.r.t prior work. The authors clarified that prior interactive CMB work concerns test-time interventions, whereas this approach is a concept-level human feedback leading to retraining. The authors clarified and added references.

- The reviewer found some definitions unclear, which were clarified.

- The lack of theoretical insights is a limitation of this work, which was acknowledged by the authors.

- Some questions on possibility to generalize this via LLMs, which was discussed as potential future work.

**Reviewer Scores:**

- Rev 16NR was initially supportive (6), and seemed relatively content to the extent of the authors' responses and rebuttals, so I anticipate they would have increased their scores to a 7.
- Rev SPcz (originally a 4) seemed satisfied with the provided answers, so I anticipate this would have improved to a 5 or 6.
- Rev AMnB (originally an 8) was supportive, and their questions were properly addressed.
- Rev jWmY (originally a 4) seemed content with the authors responses, potentially increasing their score to a 5.

---

### Decision · Program_Chairs · 2026-01-26

Accept (Poster)